# Numerical investigations on the modelling of ultrafine particles in SSH-aerosol-v1.3a: size resolution and redistribution

Oscar Jacquot[1] and Karine Sartelet[1]

[1]CEREA, Ecole des Ponts, Institut Polytechnique de Paris, EdF R&D, IPSL, Marne la Vallée, France

**Correspondence:** Oscar Jacquot (oscar.jacquot@enpc.fr) and Karine Sartelet (karine.sartelet@enpc.fr)

**Abstract.** As the health impacts of ultrafine particles become better understood, accurately modeling size distribution and number concentration in chemistry transport models is becoming increasingly important. The number concentrations is strongly affected by processes linked to aerosol dynamics: coagulation, condensation and gas/particle phase partitioning, nucleation. Coagulation is usually solved using an Eulerian approach, relying on a fixed discretization of particle sizes. In contrast, condensation and evaporation processes are rather solved using a Lagrangian approach, requiring redistribution of particles on the fixed size mesh. Here, a new analytic formulation is presented to compute efficiently coagulation partition coefficients, allowing to dynamically adjust the discretization of the coagulation operator to the size mesh evolution, and therefore solve all the processes linked to aerosol dynamics with a dynamic mesh approach, avoiding the redistribution on the fixed size grid. This new approach has the advantage of reducing the numerical diffusion introduced by condensation. The significance of these effects on number concentrations is assessed in an idealized box setting, as well as over Greater Paris with the chemistry transport model Polyphemus/Polair3D coupled to the aerosol model SSH-aerosol, using different size resolution of the particle distribution.

## 1 Introduction

As ultrafine particles, i.e. particles of diameters lower than 0.1 $\mu$m, could exert different toxicity than larger particles (Ohlwein et al., 2019; Schraufnagel, 2020; Kwon et al., 2020) and represent an uncertain part in climate models (Forster et al., 2021), it is becoming increasingly important to represent them accurately in models from the indoor and local scale (Patel et al., 2021; Frohn et al., 2021) to the global scale (Leinonen et al., 2022). Those particles are characterized by low mass but high number concentrations. Therefore, integrated mass concentrations, such as $PM_{10}$ and $PM_{2.5}$, bear little information about their significance. Chemistry-transport models (CTM) are frequently used to estimate pollutant concentrations, with applications from continental and regional scales, up to the urban scale. These models can be used to assess the impact of different emission scenarios, and they have long focused on representing the mass of particles of diameters lower than 2.5 $\mu$m and 10 $\mu$m ($PM_{2.5}$ and $PM_{10}$ respectively).

Different strategies have been developed to model the aerosol size distribution, among which the most common in CTMs are the sectional approach, which represent the distribution by piecewise approximations (e.g. Gelbard et al., 1980; Debry and Sportisse, 2007) and the modal approach, which represents the size distribution as a superposition of several modes, often

log-normal ones (e.g. Whitby and McMurry, 1997; Vignati et al., 2004; Sartelet et al., 2006). Computationally competitive and accurate numerical approaches are needed to represent both mass and number concentrations with a limited number of sections or modes. The modal approach is often favored for its low computational requirements, while the sectional approach is favored for its numerical accuracy. For modeling aerosol mass concentrations, as little as three to ten sections are used (Pilinis and Seinfeld, 1988; Fast et al., 2006; Sartelet et al., 2018; Chang et al., 2021; Menut et al., 2021). However a higher number of sections may be necessary to accurately simulate particle number concentrations, as they are strongly influenced by size distribution. The number of sections used then typically range from ten (Park et al., 2024), twenty-five (Sartelet et al., 2022), thirty sections (Adams and Seinfeld, 2002) to forty-one (Patoulias et al., 2018). The use of a large number of sections in CTMs is challenging because each section can contain multiple chemical species. As a result, the number of transported compounds in the Eulerian model is equal to the number of chemical species multiplied by the number of sections.

Aerosol dynamics involve multiple processes, which are associated to exchanges between and within phases (Warren and Seinfeld, 1985). Nucleation represents gas molecules forming a stable condensed aggregate (Laaksonen et al., 1999; Vehkamaki et al., 2002). Coagulation is associated to collision of particles, which leads to the formation of larger particles. For well-mixed systems, it is described by Smoluchowski equation (v Smoluchowski, 1918). For atmospheric aerosols, Brownian motion is the main processes leading to coagulation (Fuchs, 1964). Condensation and evaporation are dual processes involving gas/particle phase partitioning governed by the gradient between the gas-phase concentration and the concentration at the surface of the particle. The Kelvin effect plays an important role on the evolution of ultrafine particles. It models the influence of the particle curvature, which increases the apparent saturation vapor pressure of chemical compounds (Thomson, 1871; Tolman, 1949), making the condensation of semi-volatile compounds more difficult and favoring their evaporation.

Condensation and evaporation behave like a transport process, moving particles within the aerosol volume space, as they grow or shrink while interacting with the gaseous phase. One of the main drawback of the classical Eulerian framework when solving advection equations is the introduction of numerical diffusion. The Lagrangian approach is often applied in that context (Neuman, 1984; Seigneur et al., 1986; Tsang and Rao, 1988; Gelbard, 1990) in an effort to alleviate the effects of numerical diffusion, which would be introduced by the numerical discretization in an Eulerian frame of reference. Using Lagrangian approach to represent the aerosol size discretization conflicts with the Eulerian framework typically chosen to solve aerosol coagulation, which relies upon a fixed discretization through time. Hence, "moving sectional" models are designed to resolve condensation and evaporation processes (Kim and Seinfeld, 1990). However, modeling coagulation is essential to represent the formation of ultrafine particles.

To solve both coagulation and condensation/evaporation, models are required to switch between Lagrangian and Eulerian frameworks, introducing numerical diffusion which may hinder numerical performance. One advantage of maintaining a fixed discretization is that it eliminates the need for rediscretizing the coagulation operator, which would otherwise require computing partition coefficients. In fact, the discretized equations governing aerosol dynamics through coagulation involve partition coefficients that account for the possibility that the coagulation of particles from two given size sections may produce particle sizes spanning multiple sections. If the size mesh remains fixed over time, these partition coefficients can be precomputed once, reused consistently, and shared across multiple trajectories. Formulations of these coefficients, such as Jacobson et al. (2005),

are based on heuristical considerations, without considering the wide range of diameters that may be encountered within a section. Other approaches (Debry and Sportisse, 2007; Dergaoui et al., 2013) are derived from assumptions on the underlying distribution of particles within each section. In Dergaoui et al. (2013); Sartelet et al. (2020), partition coefficients are estimated numerically by a Monte Carlo method, which estimates the value of integrals using a stochastic process. Although this method may be computationally expensive, it is easily extended to simulate particle population with different mixing states, which involve integrals in multiple dimensions. Here, an analytical expression is derived under the assumption of uniformly distributed particles within each section. This allows the development of a moving sectional model that can resolve all processes related to aerosol dynamics.

Three dimensional chemistry-transport or global models represent the flow of air masses using a Eulerian framework (Sartelet et al., 2018; Menut et al., 2021; Appel et al., 2021). The sections or modes need to be of distinct and fixed size ranges for numerical consistency throughout the simulations. In other words, as particles grow or shrink with condensation and evaporation in each grid cell, the bounds of the sections or modes evolve. Eventually, it is necessary to redistribute the number and mass or moments, introducing numerical errors and diffusion. Different strategies have been developed to mitigate issues arising in aerosol distribution representation. In the modal approach, modes can evolve freely over the whole size spectrum. However, modes may overlap due to aerosol dynamics, leading to numerical difficulties. Mode merging schemes have been developed to mitigate these difficulties, by merging modes that are overlapping (Whitby et al., 2002; Binkowski and Roselle, 2003). Mode merging may also be applied for each mode when the diameter of the distribution exceeds a fixed diameter (Sartelet et al., 2007). In the sectional approach, different algorithms can be used to redistribute sections onto a fixed grid. They usually conserve mass, e.g. the Euler method (Gelbard et al., 1980; Seigneur, 1982; Devilliers et al., 2013), the fixed sectional method (Gelbard, 1990; Karl et al., 2022), and often conserve both mass and number, e.g. the two-moment approach (Tzivion et al., 1987; Adams and Seinfeld, 2002) used in different aerosol models such as in MOSAIC (Zaveri et al., 2008) or the aerosol model included in the CTM PMCAMx (Patoulias et al., 2018); the moving diameter (Jacobson, 1997) used in MOSAIC (Zaveri et al., 2008), SIREAM (Debry and Sportisse, 2007), SSH-aerosol (Sartelet et al., 2020); the hybrid bin method (Chen and Lamb, 1994) used in SALSA (Kokkola et al., 2018); or the Euler-coupled algorithm (Devilliers et al., 2013) used in SSH-aerosol.

In this article, a 'dynamic mesh coagulation' algorithm is proposed and implemented in the aerosol dynamics model SSH-aerosol. Similarly to the moving sectional approach, it features a Lagrangian dynamic discretization of the aerosol size range, which evolves according to the evolution prescribed by condensation and evaporation. Coagulation is solved on the resulting dynamic mesh by use of a time-dependent discretization of Smoluchowski equation. By replacing the Eulerian approach for solving coagulation with a dynamic mesh approach, this method isolates and evaluates the impact of numerical diffusion. The proposed algorithm, which avoids redistribution when solving aerosol dynamics, is presented in section 2. A 0D validation and study of the scheme is provided in section 3. The chemistry-transport model used to assess the impact on concentrations and the setup of the 3D simulations are presented in section 4. Finally, the impact of different size resolutions and of the new algorithm are presented in section 5.

 ## 2  Lagrangian and Eulerian representation of aerosol processes

Using the sectional approach, the aerosol distribution is described using the number and mass densities integrated over different intervals. Let $\{v_i\}_{i=0,m}$ be a partitioning of the interval $[v_0, v_{\max}]$ such that $v_{i-1} < v_i$ with $v$ the aerosol volume, $n$ the aerosol number density and $q_s$ the aerosol mass density of species $s$:

$$N_i(t) = \int_{v_{i-1}}^{v_i} dv\, n(v,t) \tag{1}$$

$$Q_{i,s}(t) = \int_{v_{i-1}}^{v_i} dv\, q_s(v,t) \tag{2}$$

The general dynamics equation represents the evolution of the aerosol density under the processes of coagulation, condensation-evaporation and nucleation (Gelbard et al., 1980). Detailed expressions are recalled in Appendix A, and the discretized equations using the sectional approach in Appendix B.

### 2.1  Computation of partitioning coefficients

Partitioning coefficients emerge through the classical sectional approach. Assuming a piecewise constant distribution on each interval provides a numerical closure for equations (A5) and (A6):

$$\frac{dN_i}{dt} = \frac{1}{2} \sum_j \sum_k N_j N_k \iint dv du\, K(u, v-u) \mathbb{1}_{[v_{j-1}, v_j]}(u) \mathbb{1}_{[v_{k-1}, v_k]}(v-u)$$

$$- \sum_k N_i N_k \iint dv du\, K(v, u) \mathbb{1}_{[v_{i-1}, v_i]}(v) \mathbb{1}_{[v_{k-1}, v_k]}(u) \tag{3}$$

$$\frac{dQ_{i,s}}{dt} = \sum_j \sum_k Q_j N_k \iint dv du\, K(u, v-u) \mathbb{1}_{[v_{j-1}, v_j]}(u) \mathbb{1}_{[v_{k-1}, v_k]}(v-u)$$

$$- \sum_k Q_i N_k \iint dv du\, K(v, u) \mathbb{1}_{[v_{i-1}, v_i]}(v) \mathbb{1}_{[v_{k-1}, v_k]}(u) \tag{4}$$

with $\mathbb{1}_\Omega$ the indicator function of $\Omega$, such that $\mathbb{1}_\Omega(v) = 1$ if $v \in \Omega$ and $\mathbb{1}_\Omega(v) = 0$ if $v \notin \Omega$.

With the approximation that the kernel $K$ can be factored out and estimated by an averaged quantity over each subdomain $[v_{j-1}, v_j] \times [v_{k-1}, v_k]$, it is possible to derive an algebraically closed form for the partitioning coefficients, which are only functions of the chosen volume discretization. The double integration of piecewise constant functions leads to piecewise second-order polynomials, which only dependent on mesh nodes:

$$R^i_{jk} = r_{jk}(v_i) - r_{jk}(v_{i-1}) \tag{5}$$

$$r_{jk}(v) = \frac{1}{2} \frac{1}{v_j - v_{j-1}} \frac{1}{v_k - v_{k-1}} \times \left[ s\Big(v - (v_{j-1} + v_{k-1})\Big)^2 - s\Big(v - (v_{j-1} + v_k)\Big)^2 \right.$$

$$\left. - s\Big(v - (v_j + v_{k-1})\Big)^2 + s\Big(v - (v_j + v_k)\Big)^2 \right] \tag{6}$$

with $s$ the ramp function, defined such that $s(v) = 0$ if $v < 0$ and $s(v) = v$ if $v \geq 0$. We refer to Appendix C for a derivation of this result, and to Appendix D for an equivalent expression, less compactly written but less sensitive to numerical truncation errors due to subtraction of large numbers of similar order of magnitude. Note that a similar approach as the one derived here was followed by Debry and Sportisse (2007) to estimate partition coefficients, but a mistake led to an inaccurate reported closed form. This expression was implemented in the software SSH-aerosol, and its validity checked by comparison to a coagulation test case defined in the software (Sartelet et al., 2020) that involves partition coefficients calculated with a Monte-Carlo approach.

## 2.2 Lagrangian and Eulerian formulations of aerosol dynamics

The SSH-aerosol model (Sartelet et al., 2020) is used to solve the general dynamics equations describing aerosol evolution. Co-agulation, nucleation, condensation of extremely-low volatile organic and non-volatile compounds are solved simultaneously. The condensation/evaporation of semi-volatile aerosols is modeled using either a dynamic or a bulk equilibrium approach, assuming instantaneous thermodynamic equilibrium between the gas and bulk-aerosol phases. In the bulk approach, the size-section weighting factors depend on the ratio of the mass transfer rate in the aerosol distribution; and the Kelvin effect, which limits the condensation of those compounds on ultrafine particles, is modeled following Zhu et al. (2016). Time integration is performed using the trapezoidal rule, an explicit Runge-Kutta method of order 2, with an embedded order 1 method enabling error estimates and adaptive time stepping. For both the fixed mesh and dynamic mesh coagulation schemes, the first step consists in computing the coagulation partition coefficients, which are necessary to discretize the coagulation operator.

For the fixed mesh coagulation scheme, the evolution of particles due to coagulation is simulated using the pre-computed partition coefficients on the fixed reference grid, while condensation-evaporation are treated in a Lagrangian manner. After each time step, as the diameters of particles may have evolved due to the Lagrangian formulation of condensation, a redistribution scheme is applied, such as the moving diameter (Jacobson, 1997) or the Euler-coupled scheme (Devilliers et al., 2013). The outline of this implementation is described in Algorithm 1.

To estimate the impact of redistributing every time step onto the fixed Eulerian grid, a dynamic mesh coagulation scheme is set up for aerosol dynamics, as described in Algorithm 2. Coagulation partition coefficients are then computed at the beginning of each timestep, allowing for the size mesh to evolve. Aerosol concentrations evolve in a Lagrangian manner under both coagulation and condensation-evaporation. Contrary to the fixed mesh scheme, redistribution is not applied at the end of each timestep. Hence the sections boundaries evolve with time. A safety feature is implemented, such that if section boundaries were to cross, redistribution is applied so that the integration can be followed though on a well ordered partition of the size discretization, which is a necessary condition for partition coefficients to be well defined. Note that, to fit the framework of a 3D CTM, redistribution is always performed at the end of each 0D simulation when $t_{\mathrm{final}}$ is reached. This final time corresponds to the timestep used for process splitting in the 3D model, including transport, deposition, chemistry, and aerosol dynamics. It generally corresponds to multiple timesteps of the internal dynamics of aerosols.

---

**Algorithm 1** Fixed mesh coagulation scheme

---

Compute coagulation partition coefficients

**while** $t < t_\text{final}$ **do**

    Compute number and mass concentration evolution due to coagulation, condensation/evaporation and nucleation

    Redistribute number and mass concentrations on the fixed Eulerian grid

**end while**

---

**Algorithm 2** Dynamic mesh coagulation scheme

---

**while** $t < t_\text{final}$ **do**

    Compute coagulation partition coefficients based on current size mesh

    Compute number and mass concentration evolution due to coagulation, condensation/evaporation and nucleation

    **if** Some mesh size nodes have crossed **then**

        Redistribute number and mass concentrations on the fixed Eulerian grid

    **end if**

**end while**

Redistribute number and mass concentrations on the fixed Eulerian grid

---

## 3 Fixed and dynamic mesh schemes in a 0D box setting

This sections aims at validating and illustrating the differing behaviors of the fixed and dynamic mesh schemes. To assess the impact of the two schemes without the complexity of a 3D simulation, where numerous factors affect concentrations, an idealized 0D box setting is studied, focusing solely on aerosol dynamics processes. Furthermore, to better understand the differences between the two schemes in 3D chemistry-transport model (CTM) simulations, the dynamic mesh scheme is also used while considering the constraint of Eulerian modeling, i.e., the redistribution of diameters onto a fixed grid. Indeed, CTMs simultaneously solve air flow, the merging of air masses, as well as chemistry and aerosol dynamics. At regular time intervals, aerosol distributions within each cell are mixed with those in neighboring cells according to air motion. From a discretization perspective, a key requirement for a CTM handling air flow in an Eulerian framework is that the aerosol size mesh must be consistent across neighboring cells. Consequently, aerosol size distributions must be redistributed onto a fixed mesh.

To compare schemes, two error indicators are considered. The first indicator is the relative error on integrated aerosol number concentration, which is expressed as

$$\left| \frac{\int_{v_\text{min}}^{v_\text{max}} dv \, n(v)}{\int_{v_\text{min}}^{v_\text{max}} dv \, n_\text{ref}(v)} - 1 \right| \tag{7}$$

The second error indicator is the mean relative error on aerosol number distribution. It is expressed as

$$\int_{v_\text{min}}^{v_\text{max}} dv \left| \frac{n(v) - n_\text{ref}(v)}{n_\text{ref}(v)} \right| \tag{8}$$

Those indicators are evaluated in several size ranges: from 1 nm to 10 nm, from 10 nm to 10 $\mu$m and over the whole discretization range spanning from 1 nm to 10 $\mu$m. The mean relative error on distribution puts a larger penalty on smoothed out profiles, which might exhibit similar relative error on integrated quantities. Comparing both metrics offers valuable insight when studying the diffusivity of the different schemes.

## 3.1 Setup of the 0D simulation

The initial aerosol size distribution is chosen as a sum of three lognormal distributions, which parameters are identical to the hazy case of Seigneur et al. (1986). Particles are assumed to be made of sulfate. To favor nucleation and condensation, gaseous sulfuric acid and extremely low volatile organic compounds formed from the autoxidation of monoterpene (Chrit et al., 2017; Sartelet et al., 2020) are initialized with concentrations of $2 \cdot 10^{-2}$ $\mu$g.m$^{-3}$. Temperature is set to 27°, pressure to 1 atm and relative humidity to 40%. Simulations are performed over a one hour duration. Particles are assumed to lie within the 1 nm to 10 $\mu$m range, and several discretization levels are considered using a geometrical refinement of the mesh. The simulations are performed with different number of sections: 4, 12, 25 and 50, while a reference simulation is computed using the fixed scheme and 200 sections.

## 3.2 Comparison between fixed mesh and dynamic mesh coagulation

This section analyses the aerosol number size distribution after the 1-hour simulation using either the fixed or the dynamic mesh coagulation schemes. The size distributions are shown in Figure 1. Aerosol dynamics primarily affect distributions for diameters below 10 nm. The reference distribution, simulated with the fixed mesh scheme and a large number of sections (200), indicates that a sharp particle mode is formed within the 1-10 nm range. When a small number of section is used, the fixed mesh scheme seems to be less efficient at representing the large variations in the aerosol number distribution than the dynamic mesh scheme, and simulation results are more smoothed out.

The distributions obtained with both schemes are compared in terms of relative error against the reference simulation using 200 sections. Figure 2 shows the relative errors on integrated aerosol number concentration, while Figure 3 shows the relative errors on aerosol number distribution. For particles in the range 1 - 10 nm, the dynamic mesh scheme consistently outperforms the fixed mesh scheme, yielding lower errors for both error indicators. The difference between the two schemes is more pronounced when comparing relative errors in number distribution, rather than errors in integrated number concentrations. This suggests that the enhanced performance is due to the less smoothed aerosol distribution. For particles with diameters higher than 10 nm, both the fixed and dynamic mesh coagulation schemes produce similar errors for a given number of sections, with errors decreasing as the number of sections increases. The similarity between both schemes in this diameter range is expected, as the time evolution is much slower. However, the dynamic mesh coagulation scheme requires more computational time than the fixed mesh coagulation scheme for a given number of sections, as it necessitates frequent re-discretizations of the coagulation operator. Figures 4 and 5 show the errors as a function of execution time for different number of sections. The overall trends are similar for both schemes, with an increase in execution time and a decrease in error as the number of sections increases. For particles of diameters in the 1-10 nm range, although the dynamic scheme requires more computational

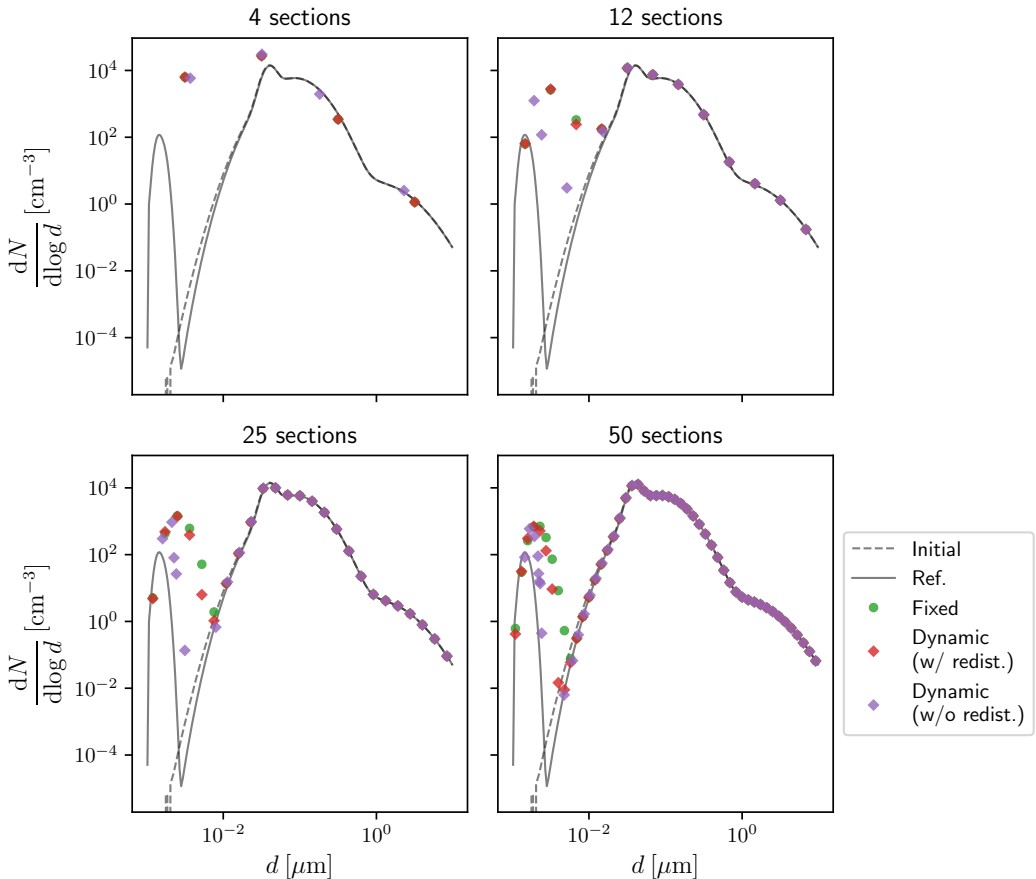

**Figure 1.** Evolution of number distribution over 1 h for different numbers of sections (4 in the upper left panel, 12 in the upper right panel, 25 in the lower left panel and 50 in the lower right panel). The reference distribution obtained with the fixed scheme and 200 sections is indicated in plain black line. The results of the fixed scheme are represented by green circles, those of the dynamic scheme by purple diamonds, and those of the dynamic scheme with regular redistribution by red diamonds.

time than the fixed scheme, it achieves lower error values, particularly in the number distribution. In contrast, the fixed scheme shows only a slow reduction in errors. For particles of diameters larger than 10 nm, both schemes yield very similar results in terms of accuracy, as there is little evolution in this size range. Consequently, the dynamic mesh is disadvantaged by its higher computation time. As a result, the curves representing the dynamic mesh scheme in Figures 4 and 5 appear as horizontal translations of those representing the fixed scheme. This highlights that the advantages of a more complex scheme are only justified in regions where aerosol dynamics are most active.

200

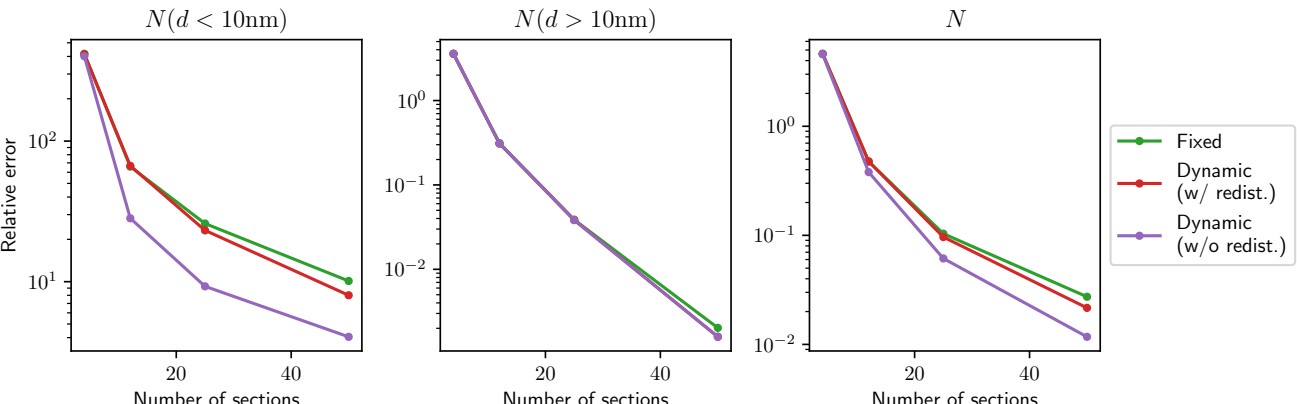

**Figure 2.** Relative error on integrated aerosol number concentration over different size ranges as a function of number of sections. Particles with diameters in the 1–10 nm range are shown in the left panel, those above 10 nm in the middle panel, and all particles in the right panel. The results of the fixed scheme are represented in green, those of the dynamic scheme in purple, and those of the dynamic scheme with regular redistribution in red.

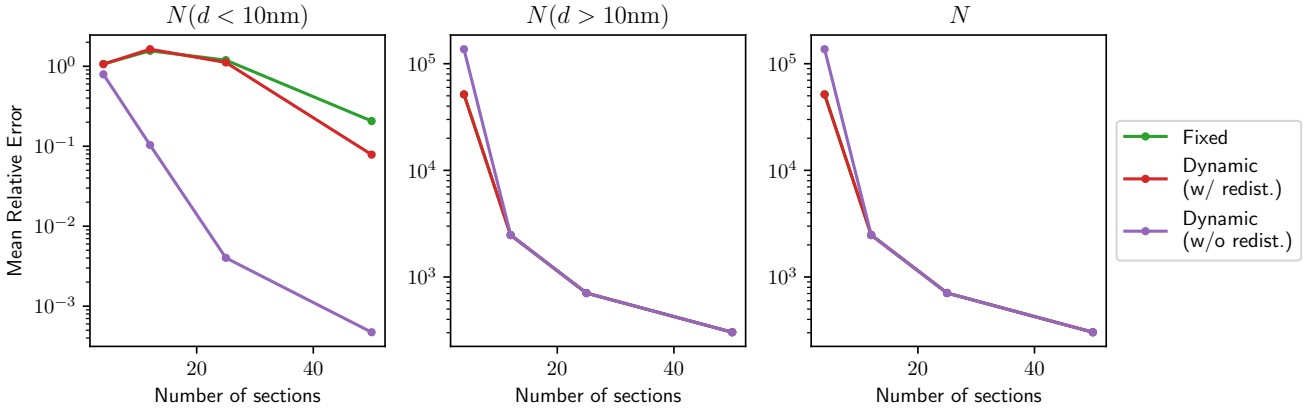

**Figure 3.** Mean relative error on aerosol number distribution over different size ranges as a function of number of sections. Particles with diameters in the 1–10 nm range are shown in the left panel, those above 10 nm in the middle panel, and all particles in the right panel. The results of the fixed scheme are represented in green, those of the dynamic scheme in purple, and those of the dynamic scheme with regular redistribution in red.

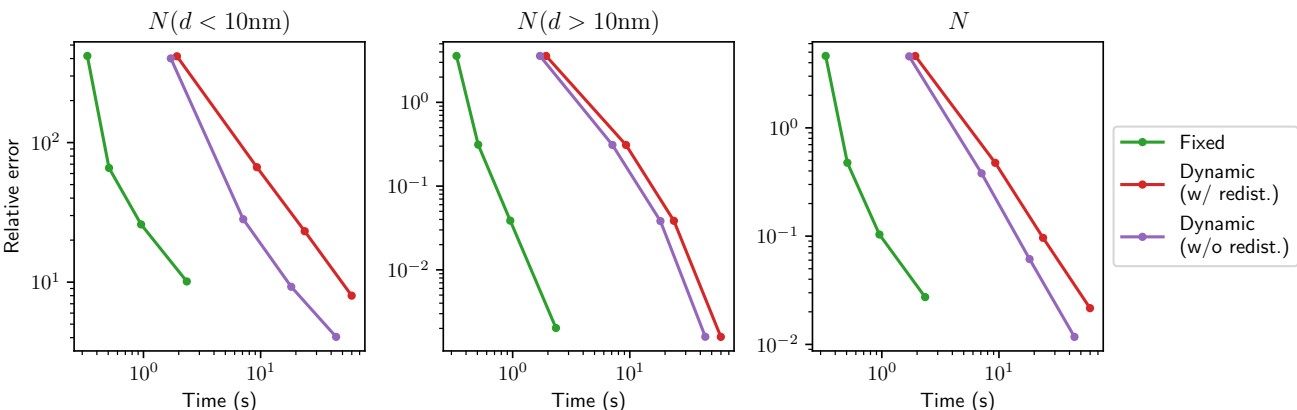

**Figure 4.** Relative error on integrated aerosol number concentration over different size ranges as a function of execution time. Particles with diameters in the 1–10 nm range are shown in the left panel, those above 10 nm in the middle panel, and all particles in the right panel. The results of the fixed scheme are represented in green, those of the dynamic scheme in purple, and those of the dynamic scheme with regular redistribution in red.

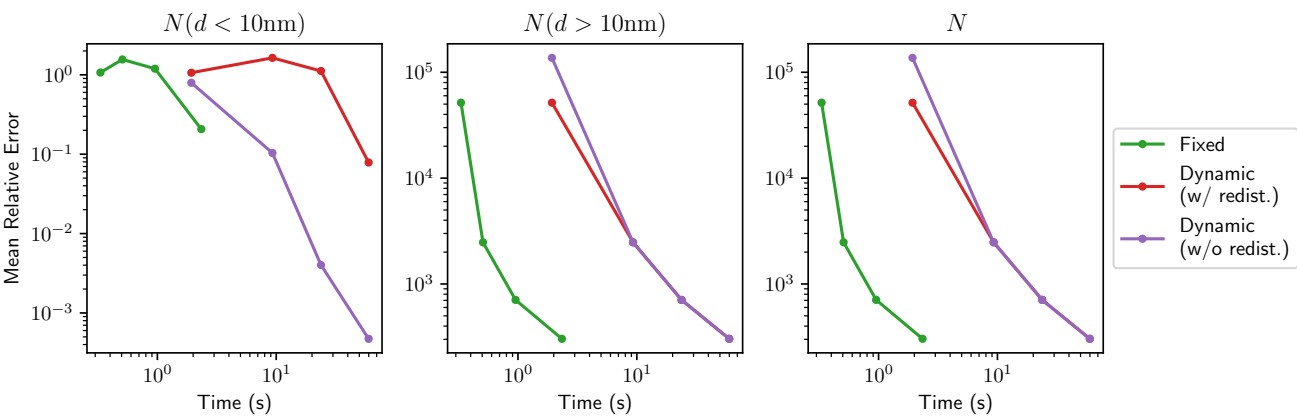

**Figure 5.** Mean relative error on aerosol number distribution over different size ranges as a function of execution time. Particles with diameters in the 1–10 nm range are shown in the left panel, those above 10 nm in the middle panel, and all particles in the right panel. The results of the fixed scheme are represented in green, those of the dynamic scheme in purple, and those of the dynamic scheme with regular redistribution in red.

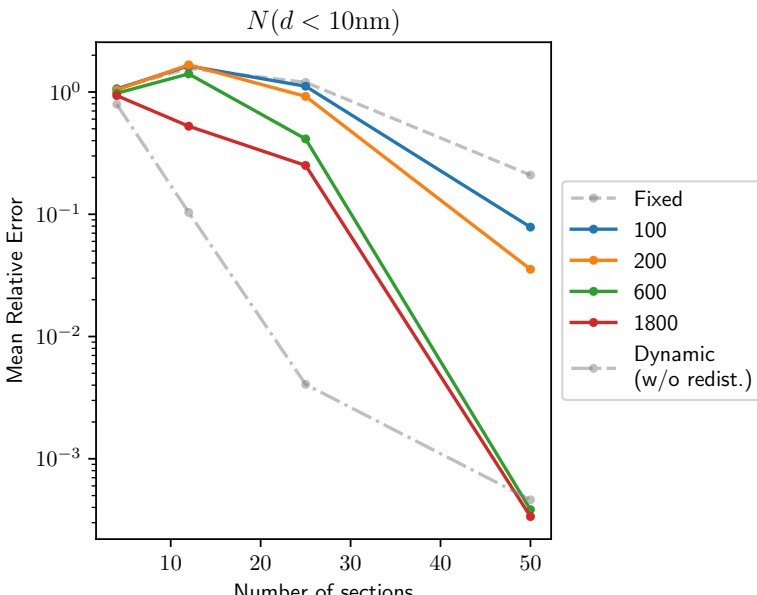

**Figure 6.** Mean relative error on aerosol number distribution for different redistribution timesteps, for particles of diameters ranging between 1 and 10 nm.

### 3.3 Dynamical mesh coagulation with redistribution at regular time intervals

In order to analyze the behavior of dynamic mesh scheme in a 3D context, one should also consider trajectories computed with the dynamic mesh with a forced redistribution step at regular time integral. In regional-scale simulations with Polyphemus/Polair3D, mixing of air masses is performed on a 100 s basis, i.e. processes related to transport, chemistry, and aerosol dynamics are split with a time step of 100 s. An intermediate scheme is added to the 0D-box comparisons. It corresponds to the dynamic mesh scheme with redistribution every 100 s, to replicate the operations performed in the 3D model.

As shown in Figures 2, 3, 4 and 5, the results of the dynamic mesh are very closed to those of the fixed mesh in terms of errors, if redistribution is applied every 100 s. In that setting, the dynamic mesh scheme loses some of its advantage, as the introduced diffusive step brings its performance closer to that of the fixed mesh scheme compared to the unperturbed dynamic mesh scheme. Figure 6 illustrates how the mean relative error evolves with different redistribution timesteps. In the limit of a large number of sections and a large redistribution timestep, the intermediate scheme behaves similarly to the dynamic mesh scheme. However, as the redistribution timestep decreases, diffusivity increases, negatively impacting the scheme's performance, making it comparable to the fixed mesh scheme but with a higher computational cost. This implies that in a 3D setting, the dynamic mesh scheme may offer similar effectiveness to the fixed mesh scheme when fluid dynamics are modeled within an Eulerian framework, depending on the number of sections and redistribution frequency. However, the dynamic mesh scheme would provide greater advantages in Lagrangian transport models.

## 4 Chemistry-transport modeling

To evaluate the impact of solving aerosol dynamics with the dynamic mesh coagulation scheme and different numbers of sections, simulations are performed over Greater Paris with the two algorithms previously described.

### 4.1 Numerical simulation setup

Numerical simulations are performed over the Greater Paris area using the Polyphemus/Polair3D (Mallet et al., 2007; Sartelet et al., 2018) chemistry-transport model coupled to the SSH-aerosol chemistry and aerosol dynamics model (Sartelet et al., 2020). For the reference simulation, a period of 12 days starting from 29 June 2009 is considered. The spatial resolution is $0.02° \times 0.02°$, and the setup is the same as in Sartelet et al. (2022). The processes related to aerosol dynamics are solved after the processes related to transport and gaseous chemistry, with a splitting time step of 100 s. It means that redistribution on the fixed mesh is performed every 100 s regardless of the algorithm used for aerosol dynamics. For aerosol-related processes, coagulation, condensation, evaporation, and heteromolecular nucleation are considered. Heteromolecular nucleation involves sulfuric acid and extremely low volatile compounds, which are formed from autoxidation of terpenes (Riccobono et al., 2014).

In order to investigate model sensitivity to size resolution, aerosol concentrations are simulated with three different particle size discretization, ranging from 1 nm to 10 $\mu$m. The finest discretization is made of 25 sections, the intermediate one of 14 sections, and the coarsest one of 9 sections. Section boundaries are defined similarly as in the study conducted by Sartelet et al. (2022) with geometrically uniform spacing below 1 $\mu$m. All discretization are identical between 1 $\mu$m and 10 $\mu$m. Figure 7 depicts discretizations considered in this study: 2, 4 and 8 sections are below 10 nm in the discretization with 9, 14 and 25 sections respectively, 2, 4 and 8 sections are respectively in the range 10-160 nm, and 2, 3 and 6 sections are between 160 nm and 1 $\mu$m.

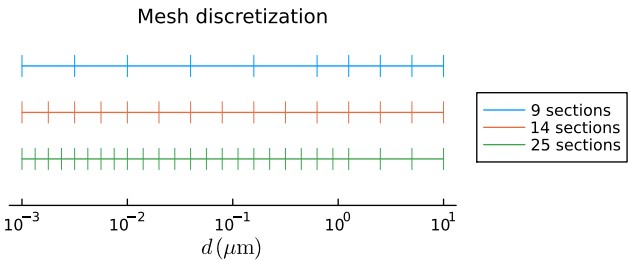

**Figure 7.** Section boundaries at each resolution level.

The redistribution method used is the Euler-Coupled algorithm (Devilliers et al., 2013). For 25 sections, emissions and boundary conditions are the same as in Sartelet et al. (2022). The consistency of these forcings across all size resolutions has been ensured by maintaining both mass and number across resolutions.

## 4.2 Comparison to observations

To assess the validity of the model, comparisons between observed and simulated concentrations are reported. Daily number concentration of particles of diameters larger than 10 nm ($N_{>10}$) are compared to measurements at two observation sites: the LHVP site (48.8°, 2.4°) representative of urban background concentrations, and the SIRTA observatory (48.7°, 2.2°), a suburban observation site. Figure 8 displays the location of the available measurements, and Figure 9 displays the simulated number concentrations over the domain considered.

Simulations performed using the dynamic mesh scheme are evaluated using multiple statistical indicators in Table 1: the Normalized Mean Error (NME), the Normalized Mean Bias (NMB) and the fraction of modeled data within a factor of 2 of observations (FAC2). Normalized mean errors and biases are similar to those presented in Sartelet et al. (2022), and are on the lower side to those simulated in different studies (Patoulias et al., 2018; Fanourgakis et al., 2019; Frohn et al., 2021; Olin et al., 2022). The FAC2 is larger than 50% for all simulations for $N_{>10}$, meeting the strictest model evaluation criterion defined in Chang and Hanna (2004). Simulations with 9, 14 and 25 sections display similar statistics for $N_{>10}$. The statistics are very similar between 9, 14 and 25 sections for $N_{>10}$, although the biases are more spread out and noticeably larger at the LHVP station when using the lowest resolution tested, being 9 sections. The simulated concentrations of $PM_{2.5}$ compare very well to the measurements, and the statistics for model to measurement comparisons of $PM_{2.5}$ are very similar between the simulations with 9, 14 and 25 sections, as shown in Table 2. The number concentrations simulated with 25 sections and the dynamic mesh scheme are shown in Fig. 9. As previously discussed in Sartelet et al. (2022), the concentrations are higher in Paris than in the suburbs. Statistics using the fixed mesh scheme are shown in Appendix, as they are very similar to those of Table 1.

**Table 1.** Comparison of simulated and measured daily number concentrations of particles $N_{>10}$ between 29 June and 10 July 2009, at the observation sites LHVP and SIRTA, using the dynamic mesh scheme. Mean observed ($\bar{o}$) and mean simulated ($\bar{s}$) daily number concentrations are reported in $\#.cm^{-3}$. Fraction of modeled data within a factor of 2 of observations (FAC2) as well as normalized mean bias (NMB) and normalized mean error (NME) are reported in %.

| Statistical indicator | SIRTA | | | | | LHVP | | | | |
| --- | --- | --- | --- | --- | --- | --- | --- | --- | --- | --- |
| | $\bar{o}$ | $\bar{s}$ | FAC2 | NMB | NME | $\bar{o}$ | $\bar{s}$ | FAC2 | NMB | NME |
| Unit | $(\#.cm^{-3})$ | $(\#.cm^{-3})$ | (%) | (%) | (%) | $(\#.cm^{-3})$ | $(\#.cm^{-3})$ | (%) | (%) | (%) |
| 9 sections | 5215 | 4766 | 75 | -9 | 36 | 8804 | 7104 | 92 | -19 | 30 |
| 14 sections | 5215 | 5444 | 92 | 4 | 36 | 8804 | 8231 | 99 | -7 | 29 |
| 25 sections | 5215 | 5322 | 92 | 2 | 35 | 8804 | 8285 | 99 | -6 | 28 |

## 5 Influence of the size resolution and redistribution

Model output sensitivity to numerical diffusion is estimated by comparing the dynamic mesh algorithm to the standard fixed mesh one. The sensitivity to the grid resolution is also studied, and provides valuable information about the ability to reduce

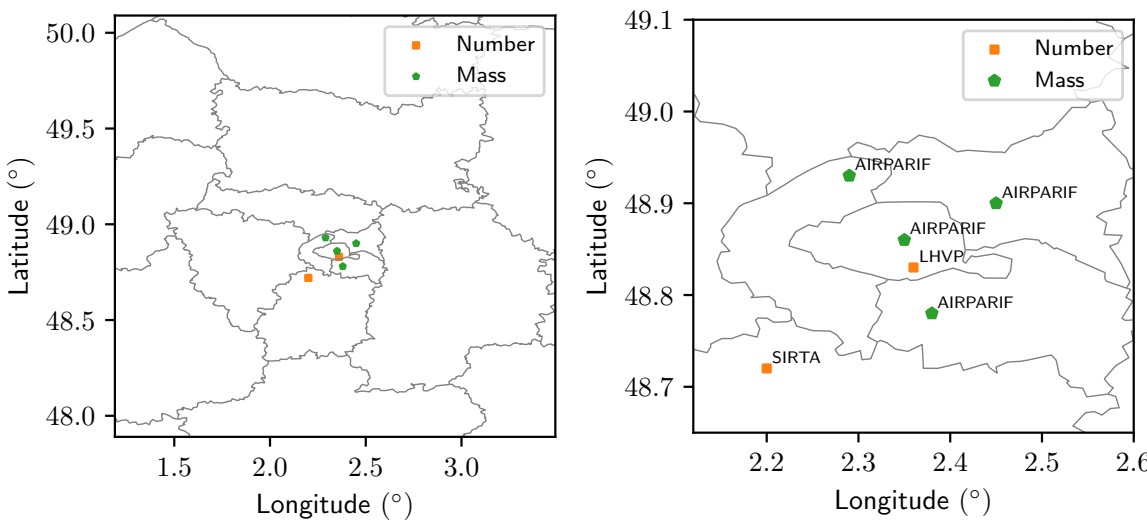

**Figure 8.** Location of observation sites, for reported number and mass measurements. Left panel represents the whole domain considered, right panel represents the area nearest to Paris where most observation sites are concentrated. For geographical context, background lines indicate borders of administrative departments around Paris area, the most central one indicating the city of Paris.

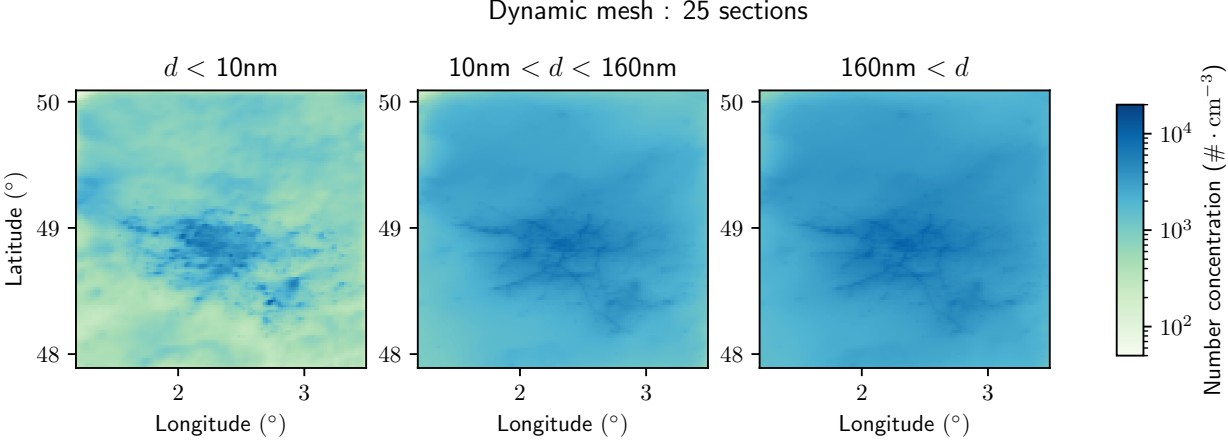

**Figure 9.** Aerosol number concentrations simulated with 25 sections and the dynamic mesh algorithm

**Table 2.** Comparison of simulated and measured daily $PM_{2.5}$ concentrations between 29 June and 10 July 2009, at four available measurement stations available from the AIRPARIF network, using the dynamic mesh scheme. Mean observed ($\bar{o}$) and mean simulated ($\bar{s}$) daily mass concentrations are reported in $\mu g.m^{-3}$. Fraction of modeled data within a factor of 2 of observations (FAC2) as well as normalized mean bias (NMB) and normalized mean error (NME) are reported in %.

| Statistical indicator | $\bar{o}$ | $\bar{s}$ | FAC2 | NMB | NME |
|:---:|:---:|:---:|:---:|:---:|:---:|
| Unit | $(\mu g.m^{-3})$ | $(\mu g.m^{-3})$ | (%) | (%) | (%) |
| 9 sections | 10.4 | 8.7 | 94 | -10 | 32 |
| 14 sections | 10.4 | 8.9 | 94 | -8 | 31 |
| 25 sections | 10.4 | 9.0 | 94 | -8 | 30 |

numerical diffusion by increasing resolution in an Eulerian setting, as well as an estimation of the relative magnitude of numerical errors associated to numerical diffusion and other error sources.

### 5.1 Sensitivity to numerical diffusion

The simulations using the fixed and dynamic mesh schemes are compared using either 9, 14 or 25 sections in Fig. 10, 11 and 12 respectively, and time-space averages are compiled in Table 3. The comparison is performed for the number of particles of diameters lower than 10 nm ($N_{<10}$), between 10 nm and 160 nm ($N_{10-160}$), and higher than 160 nm ($N_{>160}$). For each size resolution considered, average relative differences between the number concentrations simulated with both algorithms are higher for particles of smaller diameters: they are higher for $N_{<10}$ than for $N_{10-160}$ than for $N_{>160}$. This is consistent with

the expected properties of particles, as small particles are more influenced by aerosol dynamics and evolve more quickly than large particles. They are therefore the one most susceptible to numerical diffusion.

With 9 and 14 sections, the average relative differences for $N_{<10}$ between simulations using the fixed and dynamic mesh schemes are about 16% and 5% respectively (Table 3). They can be much higher locally, reaching 20% (Fig. 10 and 11), although the largest differences are observed where the number concentrations are lowest (Fig. 9). Relative differences are more

smoothly spatially distributed for larger particles, with relative differences staying below a few percents. The total number of particles with diameters higher than 160 nm is much less sensitive to the choice of algorithm, with relative differences around 2 to 3% on average.

At higher resolution, with 25 sections, the same general trends are observed (Table 3). While $N_{<10}$ concentrations are more sensitive to the choice of the algorithm than those of particles with higher diameters, the relative error is contained under 10%

globally (Figure 12), and at 3.3% on average. Compared to 9 and 14 sections, concentrations are less sensitive to the choice of the algorithm. This is an expected behavior, as higher resolution fixed mesh schemes are themselves less diffusive. At all resolution, the sensitivity of $N_{>10}$ to the choice of the algorithm is limited: 2.6% in average for 14 and 25 sections, and 3.5% for 9 sections.

**Table 3.** Average relative differences between simulations using the fixed and dynamic mesh schemes, for aerosol number concentrations. Averages are estimated over all timesteps and spatial grid points.

| Resolution | Average relative difference (%) | | | |
| --- | --- | --- | --- | --- |
| | $d < 10$ nm | $10$ nm $< d < 160$ nm | $160$ nm $< d$ | $10$ nm $< d$ |
| 9 sections | 15.8 | 4.9 | 2.5 | 3.5 |
| 14 sections | 5.4 | 3.5 | 2.0 | 2.6 |
| 25 sections | 3.3 | 3.2 | 2.2 | 2.6 |

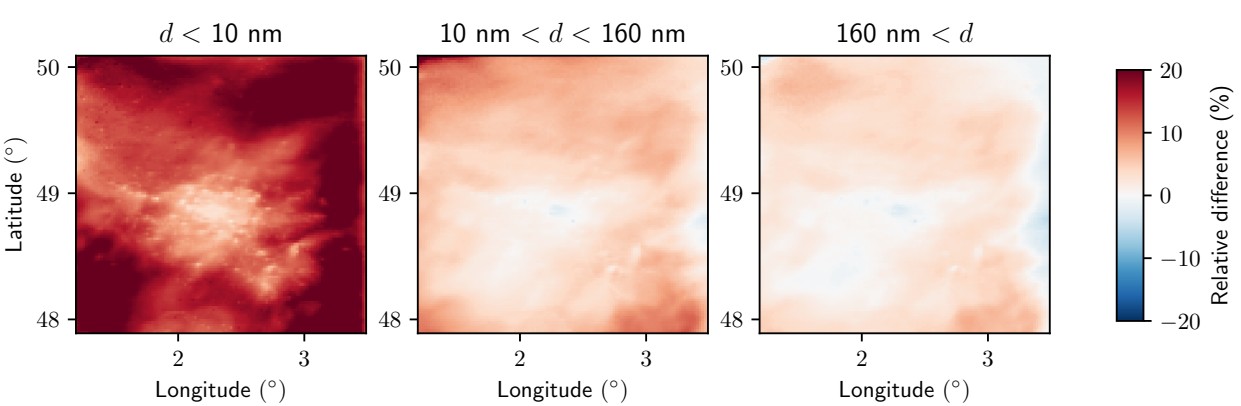

**Figure 10.** Relative difference between number concentrations simulated with the fixed and dynamic mesh schemes, using 9 sections.

## 5.2 Sensitivity to size resolution

To put into perspective the relative differences observed between the numerical algorithms, comparison is performed between the three different resolutions (9, 14 and 25 size sections). Relative differences between number concentrations for different particle diameter ranges simulated with 9 and 14 sections, using 25 sections as a reference, are displayed in Fig. 13 and 14 respectively.

   Globally, the sensitivity to the size resolution is higher than the sensitivity to the choice of the aerosol dynamics algorithm.
The $N_{<10}$ concentrations display significant variability, with average relative differences of the order of 300% for 9 sections, and 50% for 14 sections (Table 4). The sensitivity to the size resolution is lower for number concentrations of particles of higher diameters ($N_{10-160}$ and $N_{>160}$). For $N_{>10}$, the average difference between 14 and 25 sections is low (about 2.3%), but the difference between 9 and 25 sections is much higher (22%). As for the evaluation of the sensitivity to the aerosol dynamics algorithm, spatial inhomogeneities are larger for smaller particles ($N_{<10}$). The sensitivity to the size resolution is

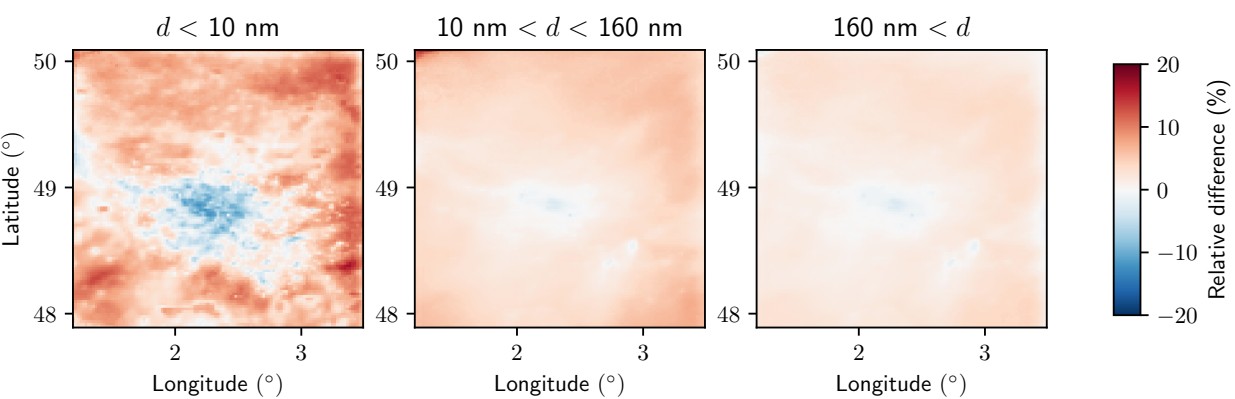

**Figure 11.** Relative difference between number concentrations simulated with the fixed and dynamic mesh schemes, using 14 sections.

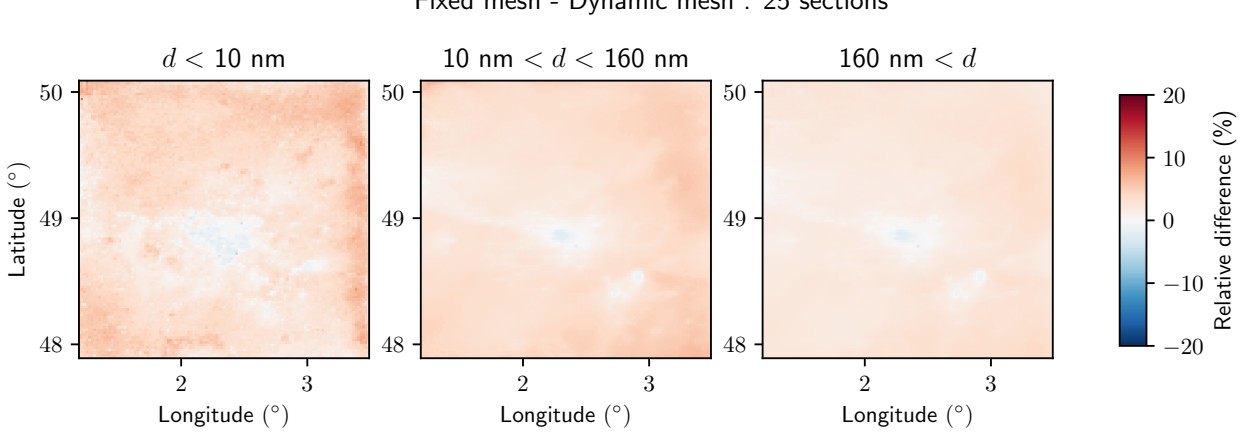

**Figure 12.** Relative difference number concentrations simulated with the fixed and dynamic mesh schemes, using 25 sections.

very similar for both schemes (Table 4). Additional figures describing the sensitivity to the size resolution using the dynamic mesh algorithm are shown in Appendix F.

## 6 Conclusions

A new algorithm that enables coupled integration of aerosol condensation, evaporation nucleation and coagulation with a dynamic particle-size mesh has been introduced. This algorithm is an extension of classical schemes for which the coagulation

**Table 4.** Average relative differences between simulations with 14 and 25 size sections, using either the fixed and dynamic mesh algorithms, for aerosol number concentration. The average is estimated over all timesteps and spatial grid points.

| Resolution | Algorithm | Average relative difference (%) compared to 25 sections | | | |
|---|---|---|---|---|---|
| | | $d < 10$ nm | $10$ nm $< d < 160$ nm | $160$ nm $< d$ | $10$ nm $< d$ |
| 9 sections | Fixed mesh | 336.3 | 15.5 | 27.3 | 22.0 |
| | Dynamic mesh | 288.7 | 16.6 | 27.4 | 22.6 |
| 14 sections | Fixed mesh | 51.8 | 10.2 | 5.0 | 2.3 |
| | Dynamic mesh | 49.9 | 10.0 | 4.7 | 2.5 |

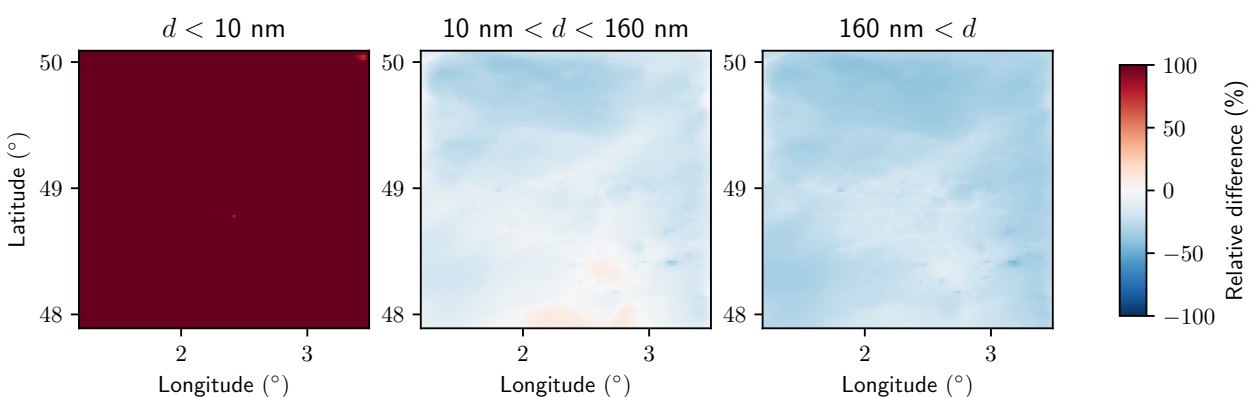

Fixed mesh : 9 - 25 sections

**Figure 13.** Relative differences between number concentrations for different particle diameter ranges, simulated with 9 and 25 sections. The fixed mesh algorithm is used.

operator is dynamically updated to match the size mesh evolution under the condensation-evaporation process. The main advantage of this scheme is to limit numerical diffusion during the resolution of aerosol dynamics.

The impact of this algorithm on the number concentrations simulated over Greater Paris was investigated with the chemistry transport model Polyphemus/Polair3D. The number concentrations of particles of diameters below 10 nm are more impacted than larger particles, as these small particles are more subject to processes linked to aerosol dynamics. The impact of the dynamic mesh algorithm decreases as the size resolution increases. It is higher when 9 size sections are used to discretize the range of diameters, than when 14 or 25 sections are used. For particles of diameters below 10 nm, the average relative difference between concentrations simulated using the fixed and dynamic mesh algorithms is about 16% with 9 sections, but only 5% with 14 sections and 3% with 25 sections. As the use of the dynamic mesh algorithm results in additional computation time, it is more relevant at low resolutions as higher benefits are then expected.

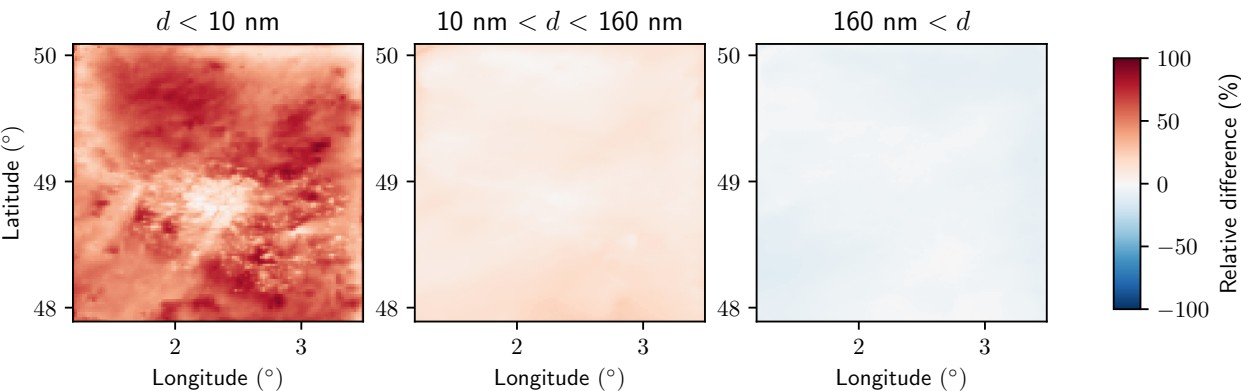

**Figure 14.** Relative differences between number concentrations for different particle diameter ranges, simulated with 14 and 25 sections. The Eulerian algorithm is used.

Number concentrations are more sensitive to the size resolution than to the aerosol dynamics algorithm, especially for the number of particles below 10 nm, indicating that averaging over wide size ranges is a limiting factor. The average differences of number concentrations for particles of diameter higher than 10 nm, computed with the finest resolution simulation as a reference, are of the order of magnitude of 20 % using 9 sections and 2 % using 14 sections. Both simulated PM$_{2.5}$ and N$_{>10}$ concentrations compare well to observations for 9, 14 and 25 sections. However, the bias of N$_{>10}$ concentrations compared to measurements is noticeably higher in the station in Central Paris for 9 than for 14 and 25 sections (-20% against -7%). Hence, 14 sections is recommended as a good compromise between complexity and performance.

The impact of the dynamical mesh algorithm on modeling number concentrations was studied with an Eulerian chemistry-transport model, requiring, for 3D consistency, regular redistribution on a fixed size mesh. However, 0D simulations have shown that the regular redistributions imposed by the assumptions of the 3D Eulerian model significantly limit the efficiency of the dynamical mesh algorithm. While in a 0D setting, this algorithm greatly reduces errors for particles strongly affected by aerosol dynamics, its advantages are diminished in the 3D Eulerian framework. Hence, it would be more suitable to use the algorithm in Lagrangian transport simulations, which deal with advection in physical space in a Lagrangian fashion (Pandis et al., 1992; Fast et al., 2012). For those types of models, regular redistribution on a fixed size grid is not needed. Therefore, one could foresee that numerical diffusion associated to the resolution of aerosol dynamics would then be the dominant source of numerical diffusion.

*Code availability.* The software code for Polyphemus/Polair3D using SSH-aerosol with Eulerian or Lagrangian coagulation is available at https://zenodo.org/doi/10.5281/zenodo.13135701, as well as the scripts to compute the statistics and graphs. The software SSH-aerosol, Polyphemus and its dependencies are distributed under the GNU General Public License v3.

## Appendix A:  General dynamics equation

Let $v$ be the aerosol volume, $n$ the aerosol number density and $q_s$ the aerosol mass density of species $s$. Under classical internal mixing assumption, which considers that aerosols of a given size are of similar chemical composition, and accounting for coagulation (coag.), condensation-evaporation (c/e) and nucleation (nucl), the evolution of the aerosol density is provided by the equation (Seinfeld and Pandis, 2012)

$$\frac{\partial n}{\partial t}(v,t) = \left.\frac{\partial n}{\partial t}\right|_{c/e}(v,t) + \left.\frac{\partial n}{\partial t}\right|_{coag.}(v,t) + \left.\frac{\partial n}{\partial t}\right|_{nucl.}(v,t) \tag{A1}$$

$$\frac{\partial q_s}{\partial t}(v,t) = \left.\frac{\partial q_s}{\partial t}\right|_{c/e}(v,t) + \left.\frac{\partial q_s}{\partial t}\right|_{coag.}(v,t) + \left.\frac{\partial q_s}{\partial t}\right|_{nucl.}(v,t) \tag{A2}$$

with

$$\left.\frac{\partial n}{\partial t}\right|_{c/e}(v,t) = -\frac{\partial}{\partial v}(I_0 n) \tag{A3}$$

$$\left.\frac{\partial q_s}{\partial t}\right|_{c/e}(v,t) = -\frac{\partial}{\partial v}(I_0 q_s) + I_s \rho_s n \tag{A4}$$

$$\left.\frac{\partial n}{\partial t}\right|_{coag.}(v,t) = \frac{1}{2}\int_{v_0}^{v} du\, K(u,v-u)n(u,t)n(v-u,t) - n(v,t)\int_{v_0}^{v_{max}} du\, K(v,u)n(u,t) \tag{A5}$$

$$\left.\frac{\partial q_s}{\partial t}\right|_{coag.}(v,t) = \int_{v_0}^{v} du\, K(u,v-u)q_s(u,t)n(v-u,t) - q_s(v,t)\int_{v_0}^{v_{max}} du\, K(v,u)n(u,t) \tag{A6}$$

and

$$\left.\frac{\partial n}{\partial t}\right|_{nucl.}(v,t) = \delta(v-v_0)\, J_0(t) \tag{A7}$$

$$\left.\frac{\partial q_s}{\partial t}\right|_{nucl.}(v,t) = \delta(v-v_0)\, J_0(t)v_0\rho_s \tag{A8}$$

where $v_0$ is the volume of the smallest condensed aerosol aggregate, $I_s$ is the volume growth rate related to condensation-evaporation for each species $s$, $I_0 = \sum_s I_s$ the total volume growth rate, $K$ is the coagulation kernel, $J_0$ the nucleation rate, $\rho_s$ is the density of species $s$, and $\delta$ is the Dirac distribution.

## Appendix B: Discretized aerosol dynamics

For coagulation, the time evolution of mass and number concentrations may be written as

$$\frac{dN_i}{dt} = \frac{1}{2} \sum_j \sum_k R^i_{jk} K_{jk} N_j N_k - N_i \sum_k K_{ik} N_k \tag{B1}$$

$$\frac{dQ^s_i}{dt} = \sum_j \sum_k R^i_{jk} K_{jk} Q^s_j N_k - Q^s_i \sum_k K_{ik} N_k \tag{B2}$$

where $K_{jk}$ is the coagulation kernel associated to collision of particles from section $j$ and $k$, and $R^i_{jk}$ is the partition coefficient, associated to particle gains in section $i$ from collisions of particles originating from sections $j$ and $k$. The coagulation kernel is modeled following Fuchs (1964), allowing to represent particles from the free molecular regime to the continuum one. A new and accurate algorithm to derive partitions coefficients is detailed in Section 2.1.

For condensation/evaporation and nucleation, the time evolution of mass and number concentrations may be written as

$$\frac{dN_i}{dt} = J_s \delta_{i,1} \tag{B3}$$

$$\frac{dQ^s_i}{dt} = 2\pi D_g d_p f(Kn_s, \alpha_s) \left[ C^s_g - C^s_a exp\left( \frac{4\sigma_s v_s}{RTd_p} \right) \right] + J_0 \frac{\pi}{6} d^3_p \rho_p \, \delta_{i,1} \tag{B4}$$

with $J_0$ the nucleation rate, $d_p$ and $\rho_p$ the particle wet diameter and density, $D_g$ and $C^s_g$ the molecular diffusivity in the air and the gas-phase concentration of species $s$, $f$ the Fuchs-Sutugin function, which depends on the Knudsen number of species $s$ ($Kn_s$) and on the accommodation coefficient $\alpha_s$, $C^s_a$ is the concentration at the particle surface assumed to be at local thermodynamic equilibrium with the particle composition, $\sigma_s$ and $v_s$ are the surface tension of species and molecular volume of species $s$.

## Appendix C: Partition coefficients for coagulation gains: closed form

Let $R^i_{jk}$ denote the fraction of particles of volume contained between $v_{i-1}$ and $v_i$, resulting from collisions of particles from sections $j$ and $k$:

$$R^i_{jk} = \int_{v_{i-1}}^{v_i} du \, (f_j * f_k)(u) \tag{C1}$$

with $*$ denoting the convolution product. Assuming uniform distribution within sections, we also have

$$f_j(v) = \frac{H(v - v_{j-1}) - H(v - v_j)}{v_j - v_{j-1}} \tag{C2}$$

$$f_k(v) = \frac{H(v - v_{k-1}) - H(v - v_k)}{v_k - v_{k-1}} \tag{C3}$$

with $H$ the Heaviside step function.

To derive a closed form for Eq. (C1), let first compute the derivative of the convolution product

$$\frac{d}{dv}(f_j * f_k) = f_j * \frac{df_k}{dv}$$

$$= f_j * \left[ \frac{1}{v_k - v_{k-1}} \left( \delta(u - v_{k-1}) - \delta(u - v_k) \right) \right]$$

$$= \frac{1}{v_k - v_{k-1}} \left[ f_j * \delta(u - v_{k-1}) - f_j * \delta(u - v_k) \right]$$

$$= \frac{1}{v_k - v_{k-1}} \left[ f_j(u - v_{k-1}) - f_j(u - v_k) \right]$$

$$= \frac{1}{v_j - v_{j-1}} \frac{1}{v_k - v_{k-1}} \left[ H(v - (v_{j-1} + v_{k-1})) - H(v - (v_j + v_{k-1})) \right.$$

$$\left. - H(v - (v_{j-1} + v_k)) + H(v - (v_j + v_k)) \right] \tag{C4}$$

We can then derive $f_j * f_k$ up to a constant $\kappa$

$$\left( f_j * f_k \right)(v) + \kappa = \int^v du \frac{d}{du} (f_j * f_k)$$

$$= \frac{1}{v_j - v_{j-1}} \frac{1}{v_k - v_{k-1}} \int^v du \left[ H(u - (v_{j-1} + v_{k-1})) - H(u - (v_j + v_{k-1})) \right.$$

$$\left. - H(u - (v_{j-1} + v_k)) + H(u - (v_j + v_k)) \right]$$

$$= \frac{1}{v_j - v_{j-1}} \frac{1}{v_k - v_{k-1}} \left[ s(v - (v_{j-1} + v_{k-1})) - s(v - (v_j + v_{k-1})) \right.$$

$$\left. - s(v - (v_{j-1} + v_k)) + s(v - (v_j + v_k)) \right]. \tag{C5}$$

As all terms are null at $v = 0$, $\kappa = 0$.

Finally, a closed form for Eq. (C1) may be written as:

$$R^i_{jk} = \int_{v_{i-1}}^{v_i} du \, (f_j * f_k)(u) = r_{jk}(v_i) - r_{jk}(v_{i-1}) \tag{C6}$$

with $r_{jk}$ a primitive of $f_j * f_k$

$$r_{jk}(v) = \frac{1}{2} \frac{1}{v_j - v_{j-1}} \frac{1}{v_k - v_{k-1}} \times \left[ s\left( v - (v_{j-1} + v_{k-1}) \right)^2 - s\left( v - (v_{j-1} + v_k) \right)^2 \right.$$

$$\left. - s\left( v - (v_j + v_{k-1}) \right)^2 + s\left( v - (v_j + v_k) \right)^2 \right] \tag{C7}$$

## Appendix D: Closed form with improved numerical stability

The closed form derived in Appendix C is analytically exact, but a direct numerical implementation under this form would lead to imprecise results do to a large sensitivity to numerical truncature under this form. For instance, if we take $v > v_j + v_k$,

all terms simplify to 1. However, a naive numerical approach would compute the square of all differences between $v$ and

395 quantities such as $v_j + v_k$. In this setting, we would then subtract numbers of similar magnitude, and possibly introduce significant rounding errors. The global form proposed in Appendix C is advantageous to simplify its derivation, but equivalent and more stable form exist. Therefore, a different form is proposed for numerical evaluation, where analytically equivalent forms are employed on different subintervals of the whole domain, improving numerical accuracy.

Let us define $\Delta v_j = v_j - v_{j-1}$ and $\Delta v_k = v_k - v_{k-1}$. Without loss of generality let us assume that $\Delta v_j > \Delta v_k$, up to a

400 permutation. Let us define

$$\alpha_{jk} = v_{j-1} + v_{k-1} \tag{D1}$$

$$\beta_{jk} = v_{j-1} + v_k \tag{D2}$$

$$\gamma_{jk} = v_j + v_{k-1} \tag{D3}$$

$$\delta_{jk} = v_j + v_k. \tag{D4}$$

These new variables are in increasing order $\alpha_{jk} < \beta_{jk} < \gamma_{jk} < \delta_{jk}$, and can be introduced in Equation (C7)

$$r_{jk}(v) = \frac{1}{2} \frac{1}{\Delta v_j \Delta v_k} \left[ s(v - \alpha_{jk})^2 - s(v - \beta_{jk})^2 - s(v - \gamma_{jk})^2 + s(v - \delta_{jk})^2 \right] \tag{D5}$$

Each interval defined by the partition of $[v_0, \infty]$ at points $\alpha_{jk}, \beta_{jk}, \gamma_{jk}, \delta_{jk}$ has an increasing amount of non-zero terms in this expression. Simplification between terms occur when considering the restriction to each of these subintervals.

$$\begin{cases} r_{jk}(v) = 0 & \text{if } v < \alpha_{jk} \\ r_{jk}(v) = \frac{1}{2} \frac{1}{\Delta v_j \Delta v_k} (v - \alpha_{jk})^2 & \text{if } \alpha_{jk} < v < \beta_{jk} \\ r_{jk}(v) = \frac{1}{2} \frac{\Delta v_k}{\Delta v_j} + \frac{v - \beta_{jk}}{\Delta v_j} & \text{if } \beta_{jk} < v < \gamma_{jk} \\ r_{jk}(v) = 1 - \frac{1}{2} \frac{1}{\Delta v_j \Delta v_k} (v - \delta_{jk})^2 & \text{if } \gamma_{jk} < v < \delta_{jk} \\ r_{jk}(v) = 1 & \text{if } v < \delta_{jk} \end{cases} \tag{D6}$$

**Appendix E: Model validation using the fixed mesh coagulation scheme**

Model to measurement comparison is provided also for the simulations using the fixed mesh coagulation scheme in Tables E1 and E2. The statistical indicators are similar to those obtained using the dynamic mesh coagulation scheme (Tables E1 and E2).

**Table E1.** Comparison of simulated and measured daily number concentrations of particles $N_{>10}$ between 29 June and 10 July 2009, at the observation sites LHVP and SIRTA, using the fixed mesh scheme. Mean observed ($\bar{o}$) and mean simulated ($\bar{s}$) daily number concentrations are reported in $\#.\text{cm}^{-3}$. Fraction of modeled data within a factor of 2 of observations (FAC2) as well as normalized mean bias (NMB) and normalized mean error (NME) are reported in $\%$.

| Statistical indicator | SIRTA | | | | | LHVP | | | | |
|---|---|---|---|---|---|---|---|---|---|---|
| | $\bar{o}$ | $\bar{s}$ | FAC2 | NMB | NME | $\bar{o}$ | $\bar{s}$ | FAC2 | NMB | NME |
| Unit | $(\#.\text{cm}^{-3})$ | $(\#.\text{cm}^{-3})$ | (%) | (%) | (%) | $(\#.\text{cm}^{-3})$ | $(\#.\text{cm}^{-3})$ | (%) | (%) | (%) |
| 9 sections | 5215 | 4806 | 62 | -8 | 35 | 8804 | 7045 | 99 | -20 | 30 |
| 14 sections | 5215 | 5463 | 92 | 10 | 35 | 8804 | 8144 | 99 | -7 | 28 |
| 25 sections | 5215 | 5422 | 92 | 9 | 35 | 8804 | 8225 | 99 | -7 | 28 |

**Table E2.** Comparison of simulated and measured daily $PM_{2.5}$ concentrations between 29 June and 10 July 2009, at four available measurement stations available from the AIRPARIF network, using the fixed mesh scheme. Mean observed ($\bar{o}$) and mean simulated ($\bar{s}$) daily mass concentrations are reported in $\mu\text{g}.\text{m}^{-3}$. Fraction of modeled data within a factor of 2 of observations (FAC2) as well as normalized mean bias (NMB) and normalized mean error (NME) are reported in $\%$.

| Statistical indicator | $\bar{o}$ | $\bar{s}$ | FAC2 | NMB | NME |
|---|---|---|---|---|---|
| Unit | $(\mu\text{g}.\text{m}^{-3})$ | $(\mu\text{g}.\text{m}^{-3})$ | (%) | (%) | (%) |
| 9 sections | 10.4 | 8.5 | 94 | -12 | 32 |
| 14 sections | 10.4 | 8.7 | 94 | -11 | 31 |
| 25 sections | 10.4 | 8.9 | 94 | -10 | 30 |

# Appendix F:  Additional figures

*Author contributions.*  KS and OJ participated to the conceptualization of the study. OJ set up the equations determining the partition coefficients. OJ and KS implemented the new scheme. OJ and KS performed the 0D numerical simulations, KS performed the 3D numerical simulations. KS and OJ conducted the visualization and wrote the manuscript.

*Competing interests.*  The authors declare no competing interests.

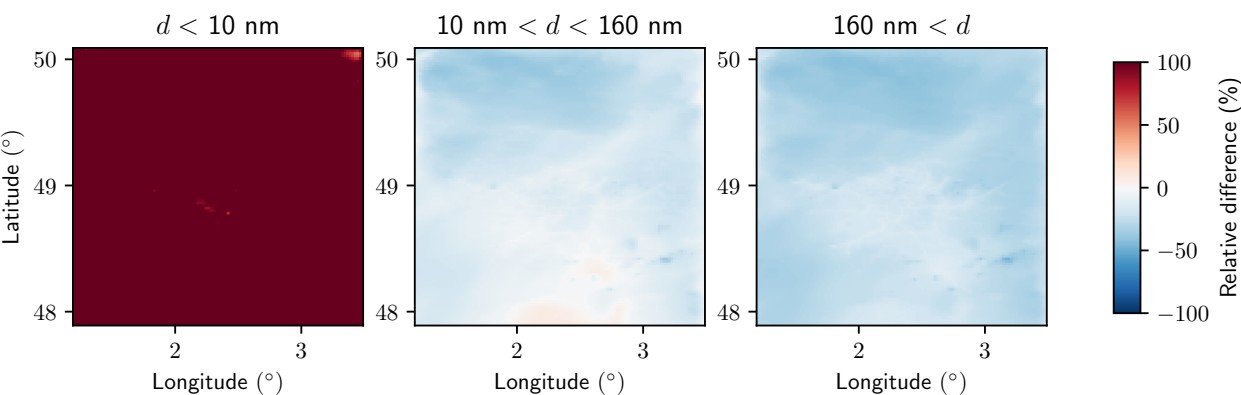

**Figure F1.** Relative difference between number concentrations for different particle diameter ranges, simulated with the coarse (9 sections) and fine discretization (25 sections), using the dynamic mesh algorithm.

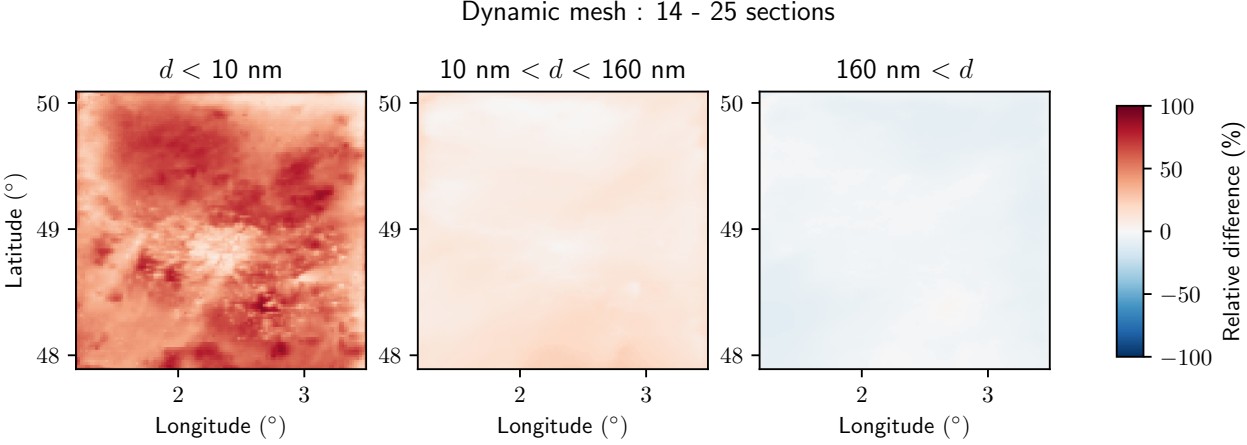

**Figure F2.** Relative difference between number concentrations for different particle diameter ranges, simulated with the medium (14 sections) and fine discretization (25 sections), using the dynamic mesh algorithm.

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
