# Peer review of "Numerical investigations on the modelling of ultrafine particles in SSH-aerosol-v1.3a: size resolution and redistribution"

_Geoscientific Model Development, 2024_

## Author Comment (AC1)

**Numerical investigations on the modelling of ultrafine particles in SSH-aerosol-v1.3a: size resolution and redistribution**

Oscar Jacquot[1] and Karine Sartelet[1]

[1]CEREA, Ecole des Ponts, Institut Polytechnique de Paris, EdF R&D, IPSL, Marne la Vallée, France

**Correspondence:** Oscar Jacquot (oscar.jacquot@enpc.fr) and Karine Sartelet (karine.sartelet@enpc.fr)

**Reply to Anonymous Referee #2's comments**

*Jacquot and Sartelet describes a semi-Lagrangian approach for representing the evolution of the aerosol size distribution through coagulation. They implemented this approach in the SSH-aerosol model and quantified the impact on simulated aerosol number concentrations over Greater Paris by coupling the updated ssh-aerosol scheme with a 3D chemical transport model. In*
5 *these simulations, they quantified the sensitivity of the simulations to the resolution of the size distribution. While the technical issues with this paper (described below) may be fixable, I am not convinced that the study addresses a critical research gap, and the technical approach is rather weak. The paper is also poorly written with many grammatical errors. Further, I do not understand why this paper is being considered for the special issue on particle-based methods; they describe an algorithm for a sectional model. I do not see how this is a particle-based method.*

10 Our reply: By using the new scheme, which is free of numerical diffusion, this work allows to provide a relative assessment of different error sources for ultrafine particle modeling. We observe that numerical diffusion is dominated by the errors related to the coarseness of the discretization used to represent the coagulation operator. To better specify the objective of this study, line 77 of the introduction

"In order to evaluate the significance of errors introduced by numerical diffusion during the coupled integration
15 of coagulation, condensation-evaporation and nucleation, a new algorithm coined 'Lagrangian aerosol dynamics' is proposed and implemented, making it possible to do away with the Eulerian approach to solve coagulation. To remain in a Lagrangian frame of reference, the representation of coagulation needs to be dynamically adapted to the size mesh evolution. The proposed algorithm, which avoids redistribution when solving aerosol dynamics, is presented in section 2. The chemistry-transport model used to assess the impact on concentrations and the setup
20 of the 3D simulations are presented in section 3. Finally, the impact of different size resolutions and of the new algorithm are presented in section 4."

was replaced by:

"A new algorithm, termed 'Lagrangian aerosol dynamics,' is proposed and implemented to enable a comparative assessment of key error sources in ultrafine particle modeling—specifically, size discretization and numerical

diffusion. By replacing the Eulerian approach for solving coagulation, this method isolates and evaluates the impact of numerical diffusion. To limit numerical diffusion and to remain in a Lagrangian frame of reference, the representation of coagulation needs to be dynamically adapted to the size mesh evolution. The proposed algorithm, which avoids redistribution when solving aerosol dynamics, is presented in section 2, along with a 0D-validation. The chemistry-transport model used to assess the impact on concentrations and the setup of the 3D simulations are presented in section 3. Finally, the impact of different size resolutions and of the new algorithm are presented in section 4."

*The paper introduces an algorithm designed to improve simulations of the aerosol size distribution, but they do not include any benchmarking for this algorithm. They show differences between their approach and a traditional sectional modeling approach, and they also quantify the impact of these differences for varying resolution of particle sizes. However, they do not show how their approach compares to analytical solutions, quantify performance against a benchmark model, or even show convergence of their solutions. They use the 25-bin sectional approach as their benchmark, but they do not show any verification of the 25-bin model. They evaluate their findings with observations, but they do not include any verification of their new algorithm against a benchmark model, which would be more appropriate.*

Our reply: We have added an additional section which provides numerical validation of the Lagrangian scheme on a benchmark in the case of an idealized 0D box model. We show the effectiveness of the scheme, which is able to provide results consistent with an Eulerian reference run at a larger resolution.

**Numerical validation**

To validate numerically the Lagrangian scheme, and to illustrate the difference between the Lagrangian and the Eulerian schemes for different size resolutions, an idealized box setting is considered. The initial mass and number distribution of particles corresponds to the sum of three lognormal distributions of the hazy case of (Seigneur et al., 1986). Particles are assumed to be made of sulfate. To favor nucleation and condensation, gaseous sulfuric acid and extremely low volatile organic compounds formed from the autoxidation of monoterpene (Chrit et al., 2017; Sartelet et al., 2020) are initialized with concentrations of $2 \cdot 10^{-2}$ $\mu$g.m$^{-3}$. Temperature is set to 27°, pressure to 1 atm and relative humidity to 40%. A one hour simulation is performed, using different size resolution levels. All gaseous species have either condensed or nucleated at the end of the simulation. This test case is highly stringent for number concentrations, as the gaseous concentrations result in intense nucleation. For each configuration, particles range from 1 nm to 10 $\mu$m and the size distribution is geometrically refined using either 4 sections, 12 sections, 25 sections or 50 sections. The reference simulation is a simulation performed with 200 sections using the Eulerian scheme.

Figure 2 and Table 2 highlight the very good agreement of both schemes on mass concentration for all size resolution. This is due to the idealized configuration of the test case, with non-volatile compounds only. The accuracy of the Eulerian scheme is nearly independent of the size resolution and relative errors reach the order

of $10 \cdot 10^{-10}$, the accuracy of the Lagrangian scheme quickly decreases from a strong relative error baseline of $10 \cdot 10^{-7}$ at only 4 sections.

[Figure]

**Figure 1.** Evolution of the number concentration simulated with the Eulerian and Lagrangian schemes at different size resolutions. The reference is computed with the Eulerian scheme using 200 sections.

For number concentrations, larger differences are observed between the schemes and the size resolution than for mass concentrations. Figure 1 and Table 1 illustrate the differences between the number concentrations simulated with the two schemes at different size resolutions. The differences with the reference simulation increase as the size resolution decreases. The differences are particularly high for 4 sections (about 460% for the number concentration), but they are much lower for 50, 25 and 12 sections (0.8%, 6% and 38% for the number concentration with the Lagrangian scheme for 50 and 25 sections respectively).

The Lagrangian scheme leads to improved accuracy, particularly for particles with a diameter lower than 10 nm. For the number concentration of particles with a diameter smaller than 10 nm, and for 12 sections and higher, the

Lagrangian scheme is able to achieve a similar accuracy to the one obtained with the Eulerian scheme using a twofold resolution. The Lagrangian scheme still outperforms the Eulerian scheme on total number concentrations, but not as strongly as for number concentrations of particles with diameter smaller than 10 nm. The trade-off to pay in terms of computational time, when choosing the Lagrangian scheme rather than the Eulerian one, is a factor of about two to three.

[Figure]

**Figure 2.** Evolution of the mass concentration simulated with the Eulerian and Lagrangian schemes at different size resolutions. The reference is computed with the Eulerian scheme using 200 sections.

Further validation is provided by the comparison to observations in a 3D context, already present in section 3.2 of the manuscript.

*The paper describes their algorithm as "Lagrangian", but they project the size distribution onto a fixed grid at every time step. As there are several truly Lagrangian aerosol and models in the literature (e.g., Shima et al., 2009 and Riemer et al.,*

| | Relative error on number concentration (absolute value) | | | | | |
|---|---|---|---|---|---|---|
| | $d \leq 10$ nm | | $d > 10$ nm | | Total number | |
| Resolution | Eulerian | Lagrangian | Eulerian | Lagrangian | Eulerian | Lagrangian |
| 4 sections | 418 | 405 | 3.58 | 3.58 | 4.61 | 4.58 |
| 12 sections | 66.5 | 28.5 | $3.12 \cdot 10^{-1}$ | $3.10 \cdot 10^{-1}$ | $4.76 \cdot 10^{-1}$ | $3.80 \cdot 10^{-1}$ |
| 25 sections | 26.3 | 9.34 | $3.84 \cdot 10^{-2}$ | $3.82 \cdot 10^{-2}$ | $1.04 \cdot 10^{-1}$ | $6.14 \cdot 10^{-2}$ |
| 50 sections | 10.3 | 2.78 | $1.87 \cdot 10^{-3}$ | $1.53 \cdot 10^{-3}$ | $2.74 \cdot 10^{-2}$ | $8.42 \cdot 10^{-3}$ |

**Table 1.** Relative error (absolute value) for different size resolution for number concentration, estimated with the Eulerian and Lagrangian schemes. The reference is computed with the Eulerian scheme using 200 sections.

| | Relative error on mass concentration (absolute value) | |
|---|---|---|
| Resolution | Eulerian | Lagrangian |
| 4 sections | $2.14 \cdot 10^{-10}$ | $1.3 \cdot 10^{-7}$ |
| 12 sections | $2.42 \cdot 10^{-10}$ | $1.45 \cdot 10^{-8}$ |
| 25 sections | $1.89 \cdot 10^{-10}$ | $6.66 \cdot 10^{-9}$ |
| 50 sections | $1.46 \cdot 10^{-10}$ | $5.10 \cdot 10^{-9}$ |

**Table 2.** Relative error (absolute value) for different size resolution for mass concentration, estimated with the Eulerian and Lagrangian schemes. The reference is computed with the Eulerian scheme using 200 sections.

*2009), I found this characterization misleading. I think it would be more accurately described as "semi-Lagrangian". I do not understand why this approach is being described in a special issue on particle-based methods.*

Our reply: Our approach is Lagrangian in aerosol volume, and would indeed be fully Lagrangian in a 0D setting where no spatial dependency is accounted for. For models which account for spatial inhomogeneity such as 3D chemistry-transport models, other processes such as advection and diffusion of air masses inevitably require that all cells share the same aerosol volume discretization. To distribute the proposed algorithm in a realistic 3D setting, a projection step to the original volume mesh is therefore necessary. However, this projection only occurs at timesteps during which neighboring cells communicate. In a 3D setting, internal dynamics in each cell is treated in a fully Lagrangian manner, between timesteps enforced by the 3D CTM.

However we do agree that our method is not particle-based, and we do not claim so. Furthermore, the referee refers to approaches which are Lagrangian in space (following the spatial evolution of each particle), but they are not Lagrangian in aerosol volume. The framework of the Smoluchowski equation describing coagulation depends on aerosol volume and time, and we studied the impact of the Lagrangian formulation in aerosol volume.

*The authors state that the numerical results are more sensitive to the resolution of the size distribution than the incorporation of their new algorithm. Even with substantial revisions, the impact of this paper as it is currently framed seems limited.*

*Perhaps it would be better to reformulate the paper to focus on the impact of the size resolution, rather than advocating for a*
95 *new algorithm that has a relatively small impact on the simulation results.*

Our reply: We aim to evaluate the relative magnitude of different error sources related to different processes. Simply increasing the resolution does not bring significative knowledge as to which potential error source dominates the remaining numerical errors. By resorting to a fully Lagrangian formulation at different resolution levels, we are able to asses the relative importance of errors due to the coarseness of the discretization of the coagulation operator compared to the numerical diffusion introduced

100 by the Eulerian formulation. For typical urban dynamics and at resolutions usual in a 3D setting, we show that numerical diffusion is not the dominant error source.

*Since the authors state that the impact of their semi-Lagrangian scheme has the greatest impact in their low-resolution simulation of 9 bins, I wonder if the proposed algorithm may be more relevant for extremely low-resolution sectional schemes*
105 *(e.g., 4 bins). WRF-Chem, for example, is often run with 4 sections.*

Our reply: In our 0D investigation, we showed that 4 sections models are generally too coarse to represent accurately aerosol dynamics for ultrafine particles. Such a low resolution is not suitable for accurate simulations where the number concentration of ultrafine particles is of interest. Our scheme enables to mitigate effects related to numerical diffusion associated to condensation-evaporation, but cannot mitigate issues related to an inaccurate representation of other processes such as coagu-

110 lation, which is inevitably a concern with such low resolution.

*The references in the introduction should be double-checked. For one thing, the aerosol chemistry model is "MOSAIC" not "MOSAIC", and the correct citation is "Zaveri, R. A., Easter, R. C., Fast, J. D., & Peters, L. K. (2008). Model for simulating aerosol interactions and chemistry (MOSAIC). Journal of Geophysical Research: Atmospheres, 113(D13)." I noticed this error*
115 *in the references because the model was misspelled, but I suggest double-checking to be sure the correct paper is referenced.*

Our reply: We have traded all occurrences of "MOZAIC" for the correct acronym "MOSAIC", and have modified the reference as suggested. Line 70

The SSH-aerosol model (Sartelet et al., 2020) is used to solve the general dynamics equations describing aerosol evolution. Coagulation, nucleation, condensation of extremely-low volatile organic and non-volatile compounds

120 are solved simultaneously. The condensation/evaporation of semi-volatile aerosols is modeled using either a dynamic or a bulk equilibrium approach, assuming instantaneous thermodynamic equilibrium between the gas and bulk-aerosol phases. In the bulk approach, the size-section weighting factors depend on the ratio of the mass transfer rate in the aerosol distribution; and the Kelvin effect, which limits the condensation of those compounds on ultrafine particles, is modeled following Zhu et al. (2016). Time integration is performed using the trapezoidal

125 rule, an explicit Runge-Kutta method of order 2, with an embedded order 1 method enabling error estimates and adaptive time stepping. For both the Eulerian and Lagrangian schemes, the first step consists in computing the coagulation partition coefficients which are necessary to discretize the coagulation operator.

For the Eulerian algorithm, the evolution of particles due to coagulation is simulated using the pre-computed partition coefficients on the fixed reference grid, while condensation-evaporation are treated in a Lagrangian manner.
After each time step, as the diameters of particles may have evolved because of the Lagrangian formulation of condensation, a redistribution scheme is applied, such as the moving diameter (Jacobson, 1997) or the Euler-coupled scheme (Devilliers et al., 2013). The outline of this implementation is described in Algorithm 1.

To estimate the impact of redistributing every time step onto the fixed Eulerian grid, a Lagrangian algorithm is setup for aerosol dynamics, as described in 2. Coagulation partition coefficients are then computed at the beginning of each timestep, allowing for the size mesh to evolve. Aerosol concentrations evolve in a Lagrangian manner under both coagulation and condensation-evaporation. Contrary to the Eulerian scheme, redistribution is not applied at the end of each timestep. Hence the sections boundaries evolve with time. A safety feature is implemented, such that if section boundaries were to cross, redistribution is applied so that the integration can be followed though on a well ordered partition of the size discretization, which is a necessary condition for partition coefficients to be well defined. Note that, to fit the framework of a 3D CTM, redistribution is always performed at the end of each 0D simulation when $t_{\text{final}}$ if reached. This final time corresponds to the timestep of th 3D-model, i.e. the time step used to solve advection and diffusion processes in space. It generally corresponds to multiple timesteps of the internal dynamics of aerosols.
* * *
**Algorithm 1** Lagrangian integration of condensation and Eulerian integration of coagulation
* * *
Compute coagulation partition coefficients

**while** $t < t_{\text{final}}$ **do**

    Compute number and mass concentration evolution due to coagulation, condensation/evaporation and nucleation

    Redistribute number and mass concentrations on the fixed Eulerian grid

**end while**
* * ** * *
**Algorithm 2** Lagrangian integration of condensation and coagulation
* * *
**while** $t < t_{\text{final}}$ **do**

    Compute coagulation partition coefficients based on current size mesh

    Compute number and mass concentration evolution due to coagulation, condensation/evaporation and nucleation

    **if** Some mesh size nodes have crossed **then**

        Redistribute number and mass concentrations on the fixed Eulerian grid

    **end if**

**end while**

Redistribute number and mass concentrations on the fixed Eulerian grid
* * *
*I suggest adding the units into the headings of Table 1 to improve readability.*

145  Our reply: For readability, we have added units to the headings of Tables 1 and 2.

**Table 3.** Comparison of simulated and measured daily number concentrations of particles $N_{>10}$ between 29 June and 10 July 2009, at the observation sites LHVP and SIRTA, using the Lagrangian scheme. Mean observed ($\bar{o}$) and mean simulated ($\bar{s}$) daily number concentrations are reported in $\#.\mathrm{cm}^{-3}$. Fraction of modeled data within a factor of 2 of observations (FAC2) as well as normalized mean bias (NMB) and normalized mean error (NME) are reported in $\%$.

| Statistical indicator | SIRTA | | | | | LHVP | | | | |
|---|---|---|---|---|---|---|---|---|---|---|
| | $\bar{o}$ | $\bar{s}$ | FAC2 | NMB | NME | $\bar{o}$ | $\bar{s}$ | FAC2 | NMB | NME |
| Unit | $(\#.\mathrm{cm}^{-3})$ | $(\#.\mathrm{cm}^{-3})$ | (%) | (%) | (%) | $(\#.\mathrm{cm}^{-3})$ | $(\#.\mathrm{cm}^{-3})$ | (%) | (%) | (%) |
| 9 sections | 5215 | 4766 | 75 | -9 | 36 | 8804 | 7104 | 92 | -19 | 30 |
| 14 sections | 5215 | 5444 | 92 | 4 | 36 | 8804 | 8231 | 99 | -7 | 29 |
| 25 sections | 5215 | 5322 | 92 | 2 | 35 | 8804 | 8285 | 99 | -6 | 28 |

**Table 4.** Comparison of simulated and measured daily $PM_{2.5}$ concentrations between 29 June and 10 July 2009, at four available measurement stations available from the AIRPARIF network, using the Lagrangian scheme. Mean observed ($\bar{o}$) and mean simulated ($\bar{s}$) daily mass concentrations are reported in $\mu\mathrm{g.m}^{-3}$. Fraction of modeled data within a factor of 2 of observations (FAC2) as well as normalized mean bias (NMB) and normalized mean error (NME) are reported in $\%$.

| Statistical indicator | $\bar{o}$ | $\bar{s}$ | FAC2 | NMB | NME |
|---|---|---|---|---|---|
| Unit | $(\mu\mathrm{g.m}^{-3})$ | $(\mu\mathrm{g.m}^{-3})$ | (%) | (%) | (%) |
| 9 sections | 10.4 | 8.7 | 94 | -10 | 32 |
| 14 sections | 10.4 | 8.9 | 94 | -8 | 31 |
| 25 sections | 10.4 | 9.0 | 94 | -8 | 30 |

*This paper contains many grammatical errors and typos. Aside from overt errors, the phrasing is often strange and unclear. The technical editing that would be required to bring this paper to a publishable form is beyond the responsibility of a peer reviewer. I strongly recommend sending this paper to a technical editor before resubmission.*

Our reply: We have carefully proofread the manuscript to remove some orthographical/grammatical errors and generally im-

150  prove the wording.

**References**

Chrit, M., Sartelet, K., Sciare, J., Pey, J., Marchand, N., Couvidat, F., Sellegri, K., and Beekmann, M.: Modelling organic aerosol concentrations and properties during ChArMEx summer campaigns of 2012 and 2013 in the western Mediterranean region, Atmos. Chem. Phys., 17, 12 509–12 531, https://doi.org/10.5194/acp-17-12509-2017, 2017.

155 Devilliers, M., Debry, É., Sartelet, K., and Seigneur, C.: A New Algorithm to Solve Condensation/Evaporation for Ultra Fine, Fine, and Coarse Particles, J. Aerosol Sci., 55, 116–136, https://doi.org/10.1016/j.jaerosci.2012.08.005, 2013.

Jacobson, M. Z.: Development and application of a new air pollution modeling system - II. Aerosol module structure and design, Atmos. Environ., 31, 131–144, https://doi.org/10.1016/1352-2310(96)00202-6, 1997.

Sartelet, K., Couvidat, F., Wang, Z., Flageul, C., and Kim, Y.: SSH-Aerosol v1.1: A Modular Box Model to Simulate the Evolution of Primary

160 and Secondary Aerosols, Atmosphere, 11, 525, https://doi.org/10.3390/atmos11050525, 2020.

Seigneur, C., Hudischewskyj, A. B., Seinfeld, J. H., Whitby, K. T., Whitby, E. R., Brock, J. R., and Barnes, H. M.: Simulation of Aerosol Dynamics: A Comparative Review of Mathematical Models, Aer. Sci. and Technol., 5, 205–222, https://doi.org/10.1080/02786828608959088, 1986.

Zhu, S., Sartelet, K. N., Healy, R. M., and Wenger, J. C.: Simulation of particle diversity and mixing state over Greater Paris: a model-

165 measurement inter-comparison, Faraday Discuss., 189, 547–566, https://doi.org/10.1039/C5FD00175G, 2016.

---

## Author Comment (AC2)

**Numerical investigations on the modelling of ultrafine particles in SSH-aerosol-v1.3a: size resolution and redistribution**

Oscar Jacquot[1] and Karine Sartelet[1]

[1]CEREA, Ecole des Ponts, Institut Polytechnique de Paris, EdF R&D, IPSL, Marne la Vallée, France

**Correspondence:** Oscar Jacquot (oscar.jacquot@enpc.fr) and Karine Sartelet (karine.sartelet@enpc.fr)

**Reply to Anonymous Referee #1's comments**

*General comments: This paper presents the development of analytical equations and a numerical algorithm for Lagrangian calculation of the coagulation term in aerosol distribution dynamics, specifically for calculating the partitioning of coagulated aerosol (number and mass) between discrete size bins in a manner consistent with the Lagrangian formulation of other aerosol dynamics processes. The approach was developed with the purpose of reducing numerical dispersion during aerosol dynamics simulation caused by redistribution due to changing between Lagrangian and Eulerian formulations. In this study, the new approach was applied within a Eulerian chemical transport model to a real-world case study, with three different resolution particle size distribution schemes. Impacts on accuracy (compared with measurements) and numerical dispersion were investigated. The main results suggest that the impact of the new formulation/algorithm on numerical diffusion is small compared with the impact of size resolution itself (and compared with the overall error in representing measurements). Overall, I find the work to be interesting and publication of the new more-internally-consistent algorithm is likely to be useful to CTM model developers. However, the impact on the field may be small due to the findings (of small effects); hence this may be better as a technical note. Additionally, some of the explanation needs improved clarity and some corrections are required, as discussed below.*

*Specific Comments: The importance and impact of this work may be limited for a few reasons, making me question whether this would be better as a technical note than a full paper. The findings indicate that numerical dispersion is less sensitive to the new coagulation formulation/algorithm than to the resolution of the aerosol size discretization. The differences are also much smaller than the simulation errors (determined by comparison to measurements).*

Our reply:

By using the new scheme, which is free of numerical diffusion, this work allows to provide a relative assessment of different error sources for ultrafine particle modeling. We observe that numerical diffusion is dominated by the errors related to the coarseness of the discretization used to represent the coagulation operator. To better specify the objective of this study, line 77 of the introduction

[revised manuscript text omitted]

*The authors state that the new formulation is basically repeating a derivation of Debry and Sportisee (2007) but correcting a mistake from that paper. This seems like a focus for a technical correction rather than for a novel contribution.*

Our reply: Our scheme provides a consistent coupling between aerosol processes in a fully Lagrangian fashion. One step of this scheme consists in an update of coagulation repartition coefficients at every timestep, to account for the evolution of the underlying mesh. This Lagrangian approach was not the focus of Debry and Sportisse (2007), who proposed a closed form for the coagulation partition coefficients regardless of coupling with other processes. During our study, we have noticed that the final closed form reported was inaccurate. Since the evaluation of a closed analytical formulation of these coefficients is key for the overall accuracy and efficiency of our scheme, we have chosen to provide the corrected closed form in Section 2.1, and added a detailed derivation in Appendix C and D.

*The authors further conclude that the new formulation/algorithm may be most useful for low size resolution simulations (9 bins), due to the computational costs of the new Lagrangian formulation/algorithm. Given this, it is not clear why one should invest (computationally) in the new formulation versus just investing in a higher resolution size distribution. A discussion of the relative computational costs should be provided.*

Our reply: The objective of the 3D investigation was assessing the relative importance of different error sources. As such, computational time was not the main focus. However, the trade-off to pay in terms of computational time, when choosing the

Lagrangian scheme rather than the Eulerian one, is a factor of about two to three.. This remains a valuable trade-off in cases such as the one displayed for the 0D validation, which is not dominated by other error sources which dilute the effectiveness of the Lagrangian scheme. The discussion of the relative computational costs is added in section 2.3.:

95     "The trade-off to pay in terms of computational time, when choosing the Lagrangian scheme rather than the Eulerian one, is a factor of about two to three."

*Tables 1 and 2 show comparison of model results to measurements for the mixed Eulerian/Lagrangian algorithm (1), not the fully Lagrangian algorithm (2). The text indicates this is because statistics "are very similar". Because the fully Lagrangian algorithm is the focus of this paper, the statistics resulting from using that algorithm should be shown (and perhaps those for* 100 *the mixed algorithm put in an appendix for comparison)*

Our reply: We have updated Tables 1 and 2 to provide validation statistics about the Lagrangian implementation in the main body of the article. We transferred the statistics about the mixed implementation in an additional Appendix, for comparison.

The Lagrangian statistics are as follows

**Table 3.** Comparison of simulated and measured daily number concentrations of particles $N_{>10}$ between 29 June and 10 July 2009, at the observation sites LHVP and SIRTA, using the Lagrangian scheme. Mean observed ($\bar{o}$) and mean simulated ($\bar{s}$) daily number concentrations are reported in #.cm$^{-3}$. Fraction of modeled data within a factor of 2 of observations (FAC2) as well as normalized mean bias (NMB) and normalized mean error (NME) are reported in %.

| | SIRTA | | | | | LHVP | | | | |
|---|---|---|---|---|---|---|---|---|---|---|
| Statistical indicator | $\bar{o}$ | $\bar{s}$ | FAC2 | NMB | NME | $\bar{o}$ | $\bar{s}$ | FAC2 | NMB | NME |
| Unit | (#.cm$^{-3}$) | (#.cm$^{-3}$) | (%) | (%) | (%) | (#.cm$^{-3}$) | (#.cm$^{-3}$) | (%) | (%) | (%) |
| 9 sections | 5215 | 4766 | 75 | -9 | 36 | 8804 | 7104 | 92 | -19 | 30 |
| 14 sections | 5215 | 5444 | 92 | 4 | 36 | 8804 | 8231 | 99 | -7 | 29 |
| 25 sections | 5215 | 5322 | 92 | 2 | 35 | 8804 | 8285 | 99 | -6 | 28 |

**Table 4.** Comparison of simulated and measured daily PM$_{2.5}$ concentrations between 29 June and 10 July 2009, at four available measurement stations available from the AIRPARIF network, using the Lagrangian scheme. Mean observed ($\bar{o}$) and mean simulated ($\bar{s}$) daily mass concentrations are reported in $\mu$g.m$^{-3}$. Fraction of modeled data within a factor of 2 of observations (FAC2) as well as normalized mean bias (NMB) and normalized mean error (NME) are reported in %.

| Statistical indicator | $\bar{o}$ | $\bar{s}$ | FAC2 | NMB | NME |
|---|---|---|---|---|---|
| Unit | ($\mu$g.m$^{-3}$) | ($\mu$g.m$^{-3}$) | (%) | (%) | (%) |
| 9 sections | 10.4 | 8.7 | 94 | -10 | 32 |
| 14 sections | 10.4 | 8.9 | 94 | -8 | 31 |
| 25 sections | 10.4 | 9.0 | 94 | -8 | 30 |

while the Eulerian statistics are given by the following tables, moved to an Appendix

**Table 5.** Comparison of simulated and measured daily number concentrations of particles $N_{>10}$ between 29 June and 10 July 2009, at the observation sites LHVP and SIRTA, using the Eulerian scheme. Mean observed ($\bar{o}$) and mean simulated ($\bar{s}$) daily number concentrations are reported in #.cm$^{-3}$. Fraction of modeled data within a factor of 2 of observations (FAC2) as well as normalized mean bias (NMB) and normalized mean error (NME) are reported in %.

| Statistical indicator | SIRTA | | | | | LHVP | | | | |
|---|---|---|---|---|---|---|---|---|---|---|
| | $\bar{o}$ | $\bar{s}$ | FAC2 | NMB | NME | $\bar{o}$ | $\bar{s}$ | FAC2 | NMB | NME |
| Unit | (#.cm$^{-3}$) | (#.cm$^{-3}$) | (%) | (%) | (%) | (#.cm$^{-3}$) | (#.cm$^{-3}$) | (%) | (%) | (%) |
| 9 sections | 5215 | 4806 | 62 | -8 | 35 | 8804 | 7045 | 99 | -20 | 30 |
| 14 sections | 5215 | 5463 | 92 | 10 | 35 | 8804 | 8144 | 99 | -7 | 28 |
| 25 sections | 5215 | 5422 | 92 | 9 | 35 | 8804 | 8225 | 99 | -7 | 28 |

**Table 6.** Comparison of simulated and measured daily PM$_{2.5}$ concentrations between 29 June and 10 July 2009, at four available measurement stations available from the AIRPARIF network, using the Eulerian scheme. Mean observed ($\bar{o}$) and mean simulated ($\bar{s}$) daily mass concentrations are reported in $\mu$g.m$^{-3}$. Fraction of modeled data within a factor of 2 of observations (FAC2) as well as normalized mean bias (NMB) and normalized mean error (NME) are reported in %.

| Statistical indicator | $\bar{o}$ | $\bar{s}$ | FAC2 | NMB | NME |
|---|---|---|---|---|---|
| Unit | ($\mu$g.m$^{-3}$) | ($\mu$g.m$^{-3}$) | (%) | (%) | (%) |
| 9 sections | 10.4 | 8.5 | 94 | -12 | 32 |
| 14 sections | 10.4 | 8.7 | 94 | -11 | 31 |
| 25 sections | 10.4 | 8.9 | 94 | -10 | 30 |

105    *Additional checking of the derivations is needed (by someone whose work involves similar derivations). I did not have the time to go through the equations in detail to fully understand them, but I did compare them to equations in the existing literature and was left with some questions. First, Appendix A lists the general aerosol dynamics equations for evolution of the number and mass distributions; a citation for this classical formulation is listed as Gelbard et al. (1980) [L91]. Although the number density equations seem generally consistent with other sources (Seinfeld and Pandis, 2006 textbook), I did not find*

110    *these formulations in the Gelbard reference, so the citation seems to need correction. More importantly, it looks like there may be a typo or error in the mass distribution equation (A4) as the units don't seem to work out. Specifically, the last term in the equation (Is\*n) appears to be missing some kind of multiplicative density term (mass of species s per volume of species s?). Without that, there is no mass, so it is unclear how that term contributes to a mass density.*

Our reply: During development, we have checked that our new scheme did provide consistent results with the previously used

115 iterative method. This method was replaced by the closed form, because of its numerical efficiency. The derivation is carried though in less than 10 successive equalities to arrive to a closed form with detailed guidance through the main steps.

We have included Seinfeld and Pandis (2012) as a more general reference which features the general dynamics equation, and included the missing specific density term in equation (A4).

$$\frac{\partial n}{\partial t}\bigg|_{c/e}(v,t) = -\frac{\partial}{\partial v}(I_0 n)$$

120
$$\frac{\partial q_s}{\partial t}\bigg|_{c/e}(v,t) = -\frac{\partial}{\partial v}(I_0 q_s) + I_s \rho_s n$$

*Additional explanation of some terms is also needed. For example, what is the meaning of the blackboard bold 1 symbol in Equations 3 and 4? Perhaps this is a well-known symbol in this subfield, but I didn't recognize it. If this work is to reach an audience that isn't excessively narrow, non-standard mathematical symbols should be defined.*

125 Our reply: We denote by $\mathbb{1}_\Omega$ the indicator function of the domain $\Omega$, meaning it has value one when evaluated within the domain and zero outside. We clarified this notation by adding a definition of this term where it is first used in the manuscript. Equations (3) and (4) are now introduced as

$$\frac{dN_i}{dt} = \frac{1}{2}\sum_j \sum_k N_j N_k \iint dv du \, K(u, v-u) \mathbb{1}_{[v_{j-1}, v_j]}(u) \mathbb{1}_{[v_{k-1}, v_k]}(v-u)$$
$$- \sum_k N_i N_k \iint dv du \, K(v, u) \mathbb{1}_{[v_{i-1}, v_i]}(v) \mathbb{1}_{[v_{k-1}, v_k]}(u)$$
$$\frac{dQ_{i,s}}{dt} = \sum_j \sum_k Q_j N_k \iint dv du \, K(u, v-u) \mathbb{1}_{[v_{j-1}, v_j]}(u) \mathbb{1}_{[v_{k-1}, v_k]}(v-u)$$
$$- \sum_k Q_i N_k \iint dv du \, K(v, u) \mathbb{1}_{[v_{i-1}, v_i]}(v) \mathbb{1}_{[v_{k-1}, v_k]}(u)$$

130 with $\mathbb{1}_\Omega$ the indicator function of $\Omega$, such that $\mathbb{1}_\Omega(v) = 1$ if $v \in \Omega$ and $\mathbb{1}_\Omega(v) = 0$ if $v \notin \Omega$.

*Table 2 caption. The units of mass concentration appear to be mislabeled here as #.cm-3, which are units for a number concentration, not a mass concentration. The units need to be corrected (or clarified).*

Our reply: We have modified the unit of mass concentration to micrograms.m-3. On Table 2, one now reads

135 *Mean observed ($\bar{o}$) and mean simulated ($\bar{s}$) daily mass concentrations are reported in $\mu g.m^{-3}$.*

*Figure 2. The figure labelling should be improved. In addition to labelling the sites based on whether they measure number or mass concentration, sites should also be labelled using the network names provided in the text (LHVP, SIRTA AIRPARIF). Additionally, labeling the location of Paris and the meaning of the polygon's boundaries would be helpful.*

Our reply: We have labeled individually each measurement station in Figure 2 on a zoomed-in version of the original map, towards the area where measurement station are concentrated. We also specified that background lines represent administrative department boundaries. Figure 2 caption now reads :

*Location of observation sites, for reported number and mass measurements. Left panel represents the whole domain considered, right panel represents the area nearest to Paris where most observation sites are concentrated. For geographical context, background lines indicate borders of administrative departments around Paris area, the most central one indicating the city of Paris.*

*Technical corrections: Figure 3. The exponent (4) is missing from the color scales and the units of number concentration are also missing (except for the exponent).*

Our reply: Thank you for pointing this out. We have also noticed additional missing characters in other figures, in labels as well as legends and graduations. After careful verification on our part, it seems that problem only came up after files were transferred to the journal. Those issues are not present in the original figures, which are also available on the Zenodo archive provided and look as intended. We will try to resolve this issue with the editor, as it seems to be purely rendering related and this aspect is not reproducible on our part.

*L20 typo: should be "focused" L42 "formerly equivalent to ...' I don't understand this. Should it say "formally"? L86. Missing word "be" before "a partitioning" L107 Typo: "substraction" L186. Should refer to Figure 6, not Figure 3.*

Our reply: We have updated the manuscript with the corrections proposed by the referee, thank you for pointing them out.

---

## Author Response (AR2)

**Numerical investigations on the modelling of ultrafine particles in SSH-aerosol-v1.3a: size resolution and redistribution**

Oscar Jacquot1 and Karine Sartelet1

1CEREA, Ecole des Ponts, Institut Polytechnique de Paris, EdF R&D, IPSL, Marne la Vallée, France **Correspondence:** Oscar Jacquot (oscar.jacquot@enpc.fr) and Karine Sartelet (karine.sartelet@enpc.fr)

**Reply to Anonymous Referee #3's comments**

I am joining the review process after the paper has undergone one round of reviews. While the revised version addresses some of the reviewers' concerns—most notably the addition of the 0D box model verification strengthens the paper—several significant weaknesses persist. These weaknesses may be challenging to resolve sufficiently to make the paper publishable. Below, I provide detailed comments regarding the methodology, nomenclature, and presentation:

- 1. Research Gap: I agree with the other reviewers that it is unclear if the paper effectively addresses a critical research gap. From the box model simulations, we learn that "the Lagrangian scheme is able to achieve a similar accuracy to the one obtained with the Eulerian scheme using a twofold resolution" (line 170), but also that "the trade-off to pay in terms of computational time, when choosing the Lagrangian scheme rather than the Eulerian one, is a factor of about two to three"
- 10 (line 171). However, the analysis lacks a cost-error plot—a standard tool for evaluating the performance of new algorithms. If my understanding of the results is correct, running the Eulerian scheme at twice the resolution would achieve a similar error at a comparable computational cost. If this is the case, it raises the question of the new algorithm's purpose and benefits. Our reply: The research gap has been better highlighted in the introduction, by adding the following sentences:

The use of a large number of sections in CTMs is challenging because each section can contain multiple chemical species. As a result, the number of transported compounds in the Eulerian model is equal to the number of chemical species multiplied by the number of sections.

•••

...

Hence, "moving sectional" models are designed to resolve condensation and evaporation processes (Kim and Seinfeld, 1990). However, modeling coagulation is essential to represent the formation of ultrafine particles.

20

15

5

Here, an analytical expression is derived under the assumption of uniformly distributed particles within each section. This allows the development of a moving sectional model that can resolve all processes related to aerosol dynamics.

The new algorithm's purpose and benefits have also been strengthened by additional analysis of the box model simulations.

- 25 To better highlight the advantages of the new scheme, two types of errors are now considered: the relative error on the integrated number concentration (as in the previous version) and the relative error on the number distribution. The second error indicator is able to penalize more significantly dynamics which are smoothed out compared to the reference, which is indicative of larger numerical diffusion. In the box-model simulation, for particles of diameters lower than 10 nm which are faster evolving, the Lagrangian scheme achieves lower errors for a given number of sections. The Eulerian scheme achieves a similar accuracy to
- 30 the one obtained with the Lagrangian scheme only for particles of diameters higher than 10 nm. When accounting for error as a function of execution time, the Lagrangian scheme is indeed penalized by its larger computation needs. Additional analysis with cost-error plots were added to section 3. The plots clearly demonstrate that the relative error of the Lagrangian scheme is significantly lower than that of the Eulerian scheme for particles with diameters below 10 nm—where aerosol dynamics are most active. Since the new scheme provides a more accurate prediction of the size distribution, the differences between the two
- 35 schemes are more pronounced for the relative error on the number distribution. The following lines have been added to section 3.1:

40

45

50

The distributions obtained with both schemes are compared in terms of relative error against the reference simulation using 200 sections. Figure 2 shows the relative errors on integrated aerosol number concentration, while Figure 3 shows the relative errors on aerosol number distribution. For particles in the range 1 - 10 nm, the dynamic mesh scheme consistently outperforms the fixed mesh scheme, yielding lower errors for both error indicators. The difference between the two schemes is more pronounced when comparing relative errors in number distribution, rather than errors in integrated number concentrations. This suggests that the enhanced performance is due to the less smoothed aerosol distribution. For particles with diameters higher than 10 nm, both the fixed and dynamic mesh coagulation schemes produce similar errors for a given number of sections, with errors decreasing as the number of sections increases. The similarity between both schemes in this diameter range is expected, as the time evolution is much slower. However, the dynamic mesh coagulation scheme requires more computational time than the fixed mesh coagulation scheme for a given number of sections, as it necessitates frequent re-discretizations of the coagulation operator. Figures 4 and 5 show the errors as a function of execution time for different number of sections. The overall trends are similar for both schemes, with an increase in execution time and a decrease in error as the number of sections increases. For particles of diameters in the 1-10 nm range, although the dynamic scheme requires more computational time than the fixed scheme, it achieves lower error values, particularly in the number distribution. In contrast, the fixed scheme shows only a slow reduction in errors. For particles of diameters larger than 10 nm, both schemes yield very similar results in terms of accuracy, as there is little evolution in this size range. Consequently, the dynamic mesh is disadvantaged by its higher computation time. As a result, the curves representing the dynamic mesh scheme in Figures 4 and 5 appear as horizontal translations of those representing the fixed scheme. This highlights that the advantages of a more complex scheme are only justified in regions where aerosol dynamics are most active.

55

2. 3D Model Implementation: The paper suggests that some remapping occurs when implementing the algorithm into the 3D model (line 180). However, the impact of this remapping on the simulation of size distributions is unclear. I recommend

60 designing a 0D test case that replicates the exact operations performed in the 3D model (e.g., redistribution every 100 s) and using this case to produce a cost-error plot. This approach would help clarify the implications of the remapping process on model performance and accuracy.

Our reply: We would like to thank you for this suggestion. We have added a new intermediate scheme in the 0D-box comparisons, which corresponds to the Lagrangian scheme with redistribution every 100 s, to replicate the operations performed in the

65 3D model. Furthermore, we have also studied the evolution of accuracy as a function of the redistribution timestep. We show that the intermediate scheme deviates from the Eulerian scheme only when the redistribution timestep or the resolution is large enough. The scheme behaves similarly to the Lagrangian in the limit of large redistribution timestep and number of sections. The following lines have been added to section 3.2:

An intermediate scheme is added to the 0D-box comparisons. It corresponds to the dynamic mesh scheme with redistribution every 100 s, to replicate the operations performed in the 3D model.

As shown in Figures 2, 3, 4 and 5, the results of the dynamic mesh are very closed to those of the fixed mesh in terms of errors, if redistribution is applied every 100 s. In that setting, the dynamic mesh scheme loses some of its advantage, as the introduced diffusive step brings its performance closer to that of the fixed mesh scheme compared to the unperturbed dynamic mesh scheme. Figure 6 illustrates how the mean relative error evolves with different redistribution timesteps. In the limit of a large number of sections and a large redistribution timestep, the intermediate scheme behaves similarly to the dynamic mesh scheme. However, as the redistribution timestep decreases, diffusivity increases, negatively impacting the scheme's performance, making it comparable to the fixed mesh scheme but with a higher computational cost. This implies that in a 3D setting, the dynamic mesh scheme may offer similar effectiveness to the fixed mesh scheme when fluid dynamics are modeled within an Eulerian framework, depending on the number of sections and redistribution frequency. However, the dynamic mesh scheme would provide greater advantages in Lagrangian transport models.

And the following lines have been added to the conclusion

However, 0D simulations have shown that the regular redistributions imposed by the assumptions of the 3D Eulerian model significantly limit the efficiency of the dynamic mesh algorithm. While in a 0D setting, this algorithm greatly reduces errors for particles strongly affected by aerosol dynamics, its advantages are diminished in the 3D Eulerian framework. Hence, it would be more suitable to use the algorithm in Lagrangian transport simulations, which deal with advection in physical space in a Lagrangian fashion.

3. Nomenclature: I agree with Reviewer 2 that the term "Lagrangian" might be misleading, as it could be confused with particle-based or super-particle methods commonly described in the literature. The term "moving sectional" model, as used

75

80

70

90 by Kim & Seinfeld (1990), may be a more accurate and appropriate descriptor for the type of model employed in this study. Our reply: We now refer to the proposed algorithm as 'dynamic mesh coagulation' and to the pre-existing method as 'fixed mesh coagulation'. This name should avoid confusion with particle-based methods by putting emphasis on the fact that our method relies on a Lagrangian description of the underlying aerosol volume mesh.

95

In this paper, the 'dynamic mesh coagulation' algorithm is proposed and implemented in the aerosol dynamics model SSH-aerosol. It features a Lagrangian dynamic discretization of the aerosol size range, which evolves in accordance to the evolution prescribed by condensation and evaporation. Coagulation is solved on the resulting dynamic mesh by use of a time-dependent discretization of Smoluchowski equation.

We would like to thank you for the comment, which helped us to better formulate the research gap question. The 'moving sectional' method developed by Kim & Seinfeld is significantly different from the method we propose, since authors clearly state that their approach is suitable when accounting only for condensation/evaporation. Our approach allows to represent both condensation/evaporation and coagulation processes under a common description, and addresses a gap in previously developed numerical methods for solving the full aerosol dynamics equation. The introduction was modified accordingly, as detailed in the reply to the first comment.

4. Writing Style: While some typos appear to have been corrected, the overall quality of the writing remains substandard
and detracts from the paper's readability. Many sentences are awkwardly phrased, such as "As the health impact of ultrafine particles is getting better understood..." and "The condensation process is formally equivalent to advection in aerosol vol-ume." A more thorough review of the language and style is necessary to meet publication standards.

Our reply:

115

120

We have improved the wording of the paper. For example, in the section where we introduce the implications of choosing 110 an Eulerian or a Lagrangian framework for condensation/evaporation and coagulation (line 42).

The condensation/evaporation process is formally equivalent to advection in aerosol volume. One of the main drawback of the classical Eulerian framework to solve advection equations is the introduction of numerical diffusion. The Lagrangian approach which aims at limiting numerical diffusion that would be introduced by the numerical discretization in an Eulerian frame of reference is therefore often applied (Neuman, 1984; Seigneur et al., 1986; Tsang and Rao, 1988; Gelbard, 1990). This Lagrangian approach is however conflicting with the Eulerian one often used to solve the coagulation process, which involves interactions between different aerosol size ranges (Gelbard et al., 1980).

Condensation and evaporation behave like a transport process, moving particles within the aerosol volume space, as they grow or shrink while interacting with the gaseous phase. One of the main drawback of the classical Eulerian framework when solving advection equations is the introduction of numerical diffusion. The Lagrangian approach is often applied in that context (Neuman, 1984; Seigneur et al., 1986; Tsang and Rao, 1988; Gelbard, 1990) in an

4

effort to alleviate the effects of numerical diffusion, which would be introduced by the numerical discretization in an Eulerian frame of reference. Using Lagrangian approach to represent the aerosol size discretization conflicts with the Eulerian framework typically chosen to solve aerosol coagulation, which relies upon a fixed discretization through time. To solve both coagulation and condensation/evaporation, models are required to switch between Lagrangian and Eulerian frameworks, introducing numerical diffusion which may hinder numerical performance.

125

In the abstract, the sentence starting by "As the health impact of ultrafine particles is getting better understood..." is replaced by

130

As the health impacts of ultrafine particles become better understood, accurately modeling size distribution and number concentration in chemistry transport models is becoming increasingly important.

**References**

135

Gelbard, F.: Modeling Multicomponent Aerosol Particle Growth By Vapor Condensation, Aer. Sci. and Technol., 12, 399–412, https://doi.org/10.1080/02786829008959355, 1990.

Gelbard, F., Tambour, Y., and Seinfeld, J. H.: Sectional Representations for Simulating Aerosol Dynamics, J. Colloid Interface Sci., 76, 541–556, https://doi.org/10.1016/0021-9797(80)90394-X, 1980.

- Kim, Y. P. and Seinfeld, J. H.: Simulation of multicomponent aerosol condensation by the moving sectional method, Journal of Colloid and Interface Science, 135, 185–199, https://doi.org/10.1016/0021-9797(90)90299-4, 1990.
  - Neuman, S. P.: Adaptive Eulerian–Lagrangian Finite Element Method for Advection–Dispersion, Int. J. Numer. Meth. Eng., 20, 321–337, https://doi.org/10.1002/nme.1620200211, 1984.
- 140 Seigneur, C., Hudischewskyj, A. B., Seinfeld, J. H., Whitby, K. T., Whitby, E. R., Brock, J. R., and Barnes, H. M.: Simulation of Aerosol Dynamics: A Comparative Review of Mathematical Models, Aer. Sci. and Technol., 5, 205–222, https://doi.org/10.1080/02786828608959088, 1986.
  - Tsang, T. H. and Rao, A.: Comparison of Different Numerical Schemes for Condensational Growth of Aerosols, Aer. Sci. and Technol., 9, 271–277, https://doi.org/10.1080/02786828808959214, 1988.